# Private Fine-tuning of Large Language Models with Zeroth-order Optimization

**Xinyu Tang**[*1]  **Ashwinee Panda**[*1]  **Milad Nasr**[2]  **Saeed Mahloujifar**[3]  **Prateek Mittal**[1]

[1]*Princeton University*  [2]*Google DeepMind*  [3]*FAIR, Meta*

**Reviewed on OpenReview:** https://openreview.net/forum?id=3Y3o0yFZfu

## Abstract

Differentially private stochastic gradient descent (DP-SGD) allows models to be trained in a privacy-preserving manner, but has proven difficult to scale to the era of foundation models. We introduce DP-ZO, a private fine-tuning method for large language models by privatizing zeroth order optimization methods. A key insight into the design of our method is that the direction of the gradient in the zeroth-order optimization we use is random and the only information from the training data is the step size, i.e., a scalar. Therefore, we only need to privatize the scalar step size, which is memory-efficient. DP-ZO provides a strong privacy-utility trade-off across different tasks, and model sizes that are comparable to DP-SGD in $(\varepsilon, \delta)$-DP. Notably, DP-ZO possesses significant advantages over DP-SGD in memory efficiency, and obtains higher utility in pure $\varepsilon$-DP when using the Laplace mechanism.

## 1 INTRODUCTION

The proliferation of open-source models pretrained on web-scale datasets (Brown et al., 2020; Zhang et al., 2022; Touvron et al., 2023) has created a paradigm shift in privacy preserving machine learning. Differential Privacy (DP) (Dwork et al., 2006) is the gold standard for preserving privacy while training models on private data, but it requires additional data (Tramèr & Boneh, 2021) to prevent a drop in utility (Yu et al., 2021a). Pretrained model checkpoints have emerged as a compelling "free" source of prior information to boost the performance of DP training (Ganesh et al., 2023; Tang et al., 2023; Panda et al., 2024a). By only requiring DP during the fine-tuning phase, a recent line of work (Li et al., 2022b;a; Yu et al., 2022; He et al., 2023; Bu et al., 2023d) is able to obtain impressive performance with

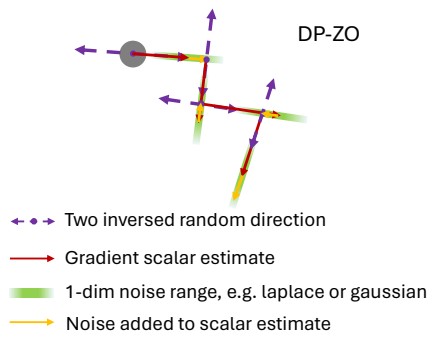

Figure 1: Visualization of DP-ZO. The only information from private data is a scalar step size for direction with lower target function value and we only need to add noise to this scalar. This scalar privatization enjoys the benefits of flexibility with DP mechanisms, ease of implementation, and reduced computation.

DP-SGD (Abadi et al., 2016). Despite these advancements, DP-SGD causes additional memory cost and needs additional engineering effort, especially for large models across devices. We propose a new direction for DP fine-tuning of large pretrained models that achieves strong privacy-utility trade-off and is more resource-efficient, easy to implement, and portable.

In this work, we study DP fine-tuning of large pretrained models with zeroth-order optimization and introduce DP-ZO. Our method uses zeroth-order optimization (ZO) (Spall, 1992). Our key insight is the synergy between differentially private fine-tuning and zeroth-order optimization. ZO provides the gradient estimates

---

[*]Equal Contribution.

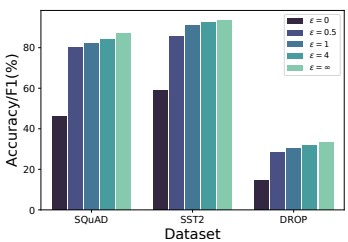

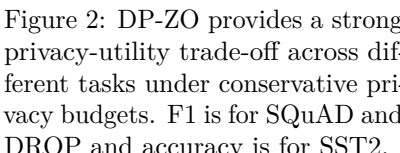

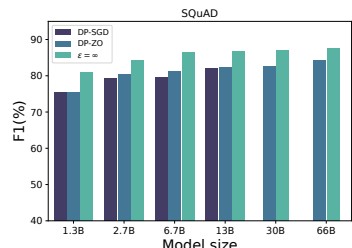

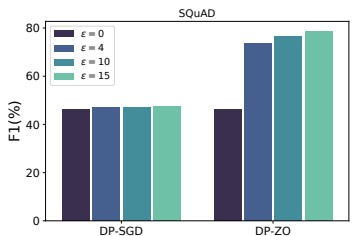

Figure 2: DP-ZO provides a strong privacy-utility trade-off across different tasks under conservative privacy budgets. F1 is for SQuAD and DROP and accuracy is for SST2.

Figure 3: DP-ZO achieves comparable performance as DP-SGD with same model size and scales seamlessly to large models like 30B/66B, that are challenging for DP-SGD.

Figure 4: DP-ZO achieves non-trivial performance for $\varepsilon$-DP. In contrast, DP-SGD (laplace) suffers to improve upon $\varepsilon = 0$ (zero-shot) due to high variance.

and the only information from private data in ZO is a scalar. We only need to privatize the scalar update by adding noise to it. Specifically, the scalar is the differences between losses from models with the same random perturbation but flipped signs. DP-ZO privatizes the zeroth-order update, by adding noise to the difference between the losses (visualized in Figure 1). This noise is proportional to the sensitivity of this loss difference with respect to changing a single example in the training set, which is controlled by clipping. We limit the $\ell_p$ sensitivity by clipping the norm of the difference in scalar losses, between the two random perturbations. Therefore, DP-ZO is flexible for both $\varepsilon$-DP and $(\varepsilon, \delta)$-DP. By removing the need for per-example gradient clipping (Abadi et al., 2016), DP-ZO enables DP training of language models with just a few lines of code without backpropagation.

DP-ZO provides a strong privacy-utility trade-off across different tasks, model sizes, dataset sizes, and DP mechanisms under conservative privacy budgets. DP-ZO only slightly degrades the performance compared to the non-private baseline (Figure 2). DP-ZO achieves comparable performance as DP-SGD within the same model size from 1.3B to 13B (Figure 3). DP-ZO scales seamlessly to large models without additional engineering, while DP-SGD requires much more memory and effort to implement per-example gradient clipping across GPUs (within a reasonable research computation limit, DP-SGD results on OPT-30B/66B are not available and omitted in Figure 3). As the model size increases to OPT-66B, the performance of DP-ZO increases and the utility gap between DP-ZO and the non-private baseline also decreases (Figure 3). Because our method only privatizes a scalar, it is compatible with multiple DP mechanisms. Specifically, DP-ZO is the first method to provide pure $\varepsilon$-DP with nontrivial utility (73.52 for SQuAD at $\varepsilon = 4$) for large models by using the Laplace mechanism (Figure 4).

Besides, we provide the empirical privacy analysis of (DP-)ZO. While ZO itself incurs less empirical privacy loss than SGD, such empirical privacy leakage estimated by membership inference attacks (Shokri et al., 2017; Panda et al., 2024b) is still much higher than random guess. Compared to ZO, DP-ZO can reduce this membership inference attack close to random guess. We also show the computation efficiency of DP-ZO over DP-SGD, even when applying gradient checkpointing and half-precision to both methods.

Independently and concurrently, Zhang et al. (2024a) also studied privatizing the scalar loss in Zeroth-order optimization with Gaussian noise and proposed DPZero. Our DP-ZO shares some similar motivation and design as DPZero, and there are several differences between our work and theirs. We provide a detailed discussion on DP-ZO and Zhang et al. (2024a) in Section 5.

## 2 BACKGROUND

### 2.1 Differential Privacy

Differential privacy (DP) is the gold standard method for providing algorithmic privacy (Dwork et al., 2006).

**Definition 2.1** $((\varepsilon, \delta)-$ Differential Privacy (DP)). We call $D, D'$ are neighboring datasets if they differ in exactly one record by adding or removing one record.[1] An algorithm $\mathcal{M}$ is said to be $(\varepsilon, \delta)$-DP if for all sets of events $S \subseteq \text{Range}(\mathcal{M})$ and neighboring datasets $D \simeq D'$ we have the guarantee:

$$\Pr[\mathcal{M}(D) \in S] \leq e^{\varepsilon} \Pr[\mathcal{M}(D') \in S] + \delta \tag{1}$$

When $\delta = 0$, we term it as pure $(\varepsilon, 0)$-DP or $\varepsilon$-DP for simplicity.

For a vector $x = (x_1, x_2, .., x_n)$, we use $l_p$ to denote the p-norm of $x$, $l_p(x) = (|x_1|^p + |x_2|^p + ... + |x_n|^p)^{\frac{1}{p}}$. We now introduce a set of existing DP mechanisms that we will use in our work.

**Proposition 2.2** (Gaussian mechanism (Dwork & Roth, 2014; Balle & Wang, 2018)). *For any function* $f : \mathbb{X}^n \to \mathbb{R}$ *with* $l_2$ *sensitivity* $\Delta$*, the mechanism defined as*

$$M(X) = f(X) + z,$$

*where* $z \sim \mathcal{N}\left(0, \sigma^2\right)$*, provides* $(\varepsilon, \delta)$*-DP where* $\Phi(\frac{\Delta}{2\sigma} - \frac{\varepsilon\sigma}{\Delta}) - e^{\varepsilon}\Phi(-\frac{\Delta}{2\sigma} - \frac{\varepsilon\sigma}{\Delta})) \leq \delta$*.* $\Phi(t)$ *is the cumulative distribution function (CDF) of the univariate Gaussian distribution* $\mathcal{N}(0, 1)$*.*

**Proposition 2.3** (Laplace mechanism (Dwork & Roth, 2014)). *For any function* $f : \mathbb{X}^n \to \mathbb{R}$ *with* $l_1$ *sensitivity* $\Delta$ *the mechanism defined as*

$$M(X) = f(X) + z,$$

*where* $z \sim Laplace\left(0, \frac{\Delta}{\varepsilon}\right)$*, provides* $(\varepsilon, 0)$*-DP.*

## 2.2 Zeroth-order Optimization

Zeroth-order optimization methods (Kiefer & Wolfowitz, 1952; Spall, 1992; Shamir, 2013; Ghadimi & Lan, 2013; Nesterov & Spokoiny, 2017) use finite difference of function values to estimate gradients, instead of computing gradients in first-order methods like SGD. By evaluating the objective function values around the points $x$, ZO provides the step size towards the direction where the point has a lower function value; See Liu et al. (2020); Zhang et al. (2024c) for a more detailed review of Zeroth-order optimization methods. Particularly, we use the difference in losses between two random perturbations SPSA (Spall, 1992; Duchi et al., 2015) with opposite signs to determine the magnitude of a gradient update in the direction of the random perturbations. Following (Spall, 1992; Malladi et al., 2023), we define SPSA in Definition 2.4.

**Definition 2.4** (Simultaneous Perturbation Stochastic Approximation (SPSA) (Spall, 1992)). Given a model with parameters $\theta \in \mathbb{R}^d$ and a loss function $\mathcal{L}$, the gradient estimate on a minibatch $\mathcal{B}$ drawn from a dataset $\mathcal{D}$ is computed by projecting the loss on the minibatch $\mathcal{L}(\theta; \mathcal{B})$ onto a random perturbation $z \in \mathbb{R}^d$ that is a standard Gaussian random variable (i.e., $z \sim \mathcal{N}(0, I_d)$) scaled by perturbation scale $\phi$:

$$\hat{\nabla}\mathcal{L}_b(\theta; \mathcal{B}) = \frac{\mathcal{L}(\theta + \phi z; \mathcal{B}) - \mathcal{L}(\theta - \phi z; \mathcal{B})}{2\phi} z \tag{2}$$

As noted in the Malladi et al. (2023), when $\phi \to 0$, the SPSA estimate could be considered as a rank-1 reconstruction of the gradient. While SPSA only provides a scalar information from the data, interestingly, Malladi et al. (2023) show this method converges at a rate that is not catastrophically slower than SGD in fine-tuning large language models in downstream tasks. Malladi et al. (2023) reason this phenomena as a result of the Hessian of the loss exhibiting small local effective rank.

Zeroth-order optimization (ZO) serves as high-variance estimates of the actual gradient (Liu et al., 2018), enabling optimization without the need for explicit gradient computations. While the update of model is from the random perturbation scaled by the step size and the only information from data is the step size, ZO still carries a privacy risk, leaking information about the data (as we show later in Section 4.3). Therefore, incorporating differential privacy into ZO is essential to safeguard against these vulnerabilities.

---

[1]This neighboring definition is for add/removal DP and the most common one. There are other neighboring definitions in the literature (Desfontaines & Pejó, 2020; Ponomareva et al., 2023).

# 3   OUR METHOD: DP-ZO

We introduce our method for differentially private zeroth order optimization (DP-ZO) by integrating DP into Definition 2.4. In DP-ZO, the information obtained from training data can be represented as a scalar. This scalar has a bounded sensitivity (when applying clipping) and can be privatized by adding noise. If we compare the noise added in DP-ZO to a single dimension to the noise added in DP-SGD to the entire gradient, we expect the univariate noise to be less detrimental to the utility (due to the curse dimensionality in differential privacy (Dwork & Roth, 2014)). In other words, we would expect the gap between non-private and private utility to be smaller than that of DP-SGD. However, it is possible that for some tasks the non-private performance of zeroth-order optimization is poor (see Section 4.3).

---

**Algorithm 1** Differentially Private-ZO

---

1: Model parameters $\theta$, dataset $D$, learning rate $\eta$, perturbation scale $\phi$, privacy parameter $\sigma$, noising mechanism $\mathcal{Z}$ (Gaussian or Laplace), clipping threshold $C$, expected batch size $B$, sub-sampling rate $p = B/|D|$).
2: **for** $t \in 1, \ldots T$ **do**
3:     Poisson sample $\mathcal{B}$ from $D$ with sub-sampling rate $p$
4:     $\vec{z} \sim \mathcal{N}(\vec{0}_{|\theta|}, \mathbf{I}_{|\theta| \times |\theta|})$
5:     $\theta^+ \leftarrow \theta + \phi\vec{z}$
6:     $\theta^- \leftarrow \theta - \phi\vec{z}$
7:     **for** $(x_i, y_i) \in \mathcal{B}$ **do**
8:         $l_i^+ \leftarrow \mathcal{L}(\theta^+, (x_i, y_i))$
9:         $l_i^- \leftarrow \mathcal{L}(\theta^-, (x_i, y_i))$
10:        $l_i = clip(l_i^+ - l_i^-, C)$
11:     **end for**
12:     **if** $\mathcal{Z}$ is Gaussian **then**
13:        $s = \dfrac{\sum_{i \in \mathcal{B}} l_i + \mathcal{N}(0, C^2\sigma^2)}{B \cdot 2\phi}$
14:     **else if** $\mathcal{Z}$ is Laplace **then**
15:        $s = \dfrac{\sum_{i \in \mathcal{B}} l_i + Laplace(0, C\sigma)}{B \cdot 2\phi}$
16:     **end if**
17:     $\theta = \theta - \eta s\vec{z}$
18: **end for**

---

**DP-ZO.** We explain the steps of our algorithm while emphasizing the key differences from Definition 2.4 required to provide DP guarantee. We first sample a batch from the dataset with Poisson sampling (Balle et al., 2018) which allows us to use privacy amplification by subsampling. For each model parameter $\theta_i$ we want to update, we independently sample a perturbation $z_i$ from a standard Gaussian distribution and scale it by a predetermined constant $\phi$; we denote the full perturbation vector as $\phi\vec{z}$. Now we compute an approximation of the gradient by projecting it onto the random perturbation $\vec{z}$. That is, for a training sample $x_i$ we compute the difference in scalar losses between $\theta + \phi\vec{z}, \theta - \phi\vec{z}$. Intuitively, this scalar tells us how much better one random step is than the other. We clip this scalar to limit the sensitivity . We add noise to the aggregation over samples in our training batch (described in detail in the subsequent paragraph). Finally, we take a step in the direction of $\vec{z}$ by scaling our private step size by the expected batch size, perturbation constant $\phi$, and the learning rate $\eta$.

**DP-ZO enables new mechanisms by privatizing the difference in losses between perturbations.** DP-ZO proposes an update direction determined by a $d$-dimensional random vector (sampled from standard Gaussian distribution) independent of the private training data. The only private information is the step size, that is influenced by the difference in losses between perturbations with opposite signs. To privatize this step size, we add noise proportional to the sensitivity of the step size. We bound the sensitivity of the step size by clipping the per-example step sizes to a specific range $[-C, C]$, so the sensitivity under add-remove DP is $C$.

Given a private scalar with bounded sensitivity, we can apply the classical Gaussian mechanism to release a privatized scalar with $(\varepsilon, \delta)$-DP. The Gaussian mechanism is widely studied in privacy-preserving machine

learning techniques like DP-SGD. However, the Gaussian mechanism can only provide $(\varepsilon, \delta)$-DP and researchers often recommend using cryptographically small values of $\delta$ (Vadhan, 2017). The stronger privacy notion, i.e., $\varepsilon$-DP, comes with a guarantee that the mechanism will never fail catastrophically. For example, we can use the Laplace mechanism to achieve the pure $\varepsilon$-DP guarantee. However, due to large tails of the Laplace mechanism, it has never been a contender for high dimensional optimization.

Although it is possible to obtain pure DP with DP-SGD by adding Laplace noise scaled to the $\ell_1$ sensitivity of the gradient, this is challenging for large models because the $\ell_1$ sensitivity can be $\sqrt{d}$ times larger than the $\ell_2$ sensitivity (and often is; see Section 4.3), especially for billion-parameter LLMs. In contrast, DP-ZO only requires privatizing the loss. The one-dimensional private estimation of the step size is amenable to the Laplace mechanism, because the $\ell_p$ norms are equivalent for the scalar. Specifically, *DP-ZO with the Laplace mechanism is the first method to achieve a reasonable privacy-utility trade-off under pure $\epsilon$-DP for private fine-tuning of LLMs.* While this work primarily explores these two mechanisms, DP-ZO is flexible enough to be extended to other differential privacy mechanisms, broadening its applicability.

**Privacy analysis.** As we consider multiple accounting methods with multiple previously proposed mechanisms, we give the overview of the analysis below and defer the full privacy analysis to Appendix A.

**Theorem 3.1.** *Algorithm 1 is $(\varepsilon, \delta)$-DP. Particularly, for Laplace mechanism $Laplace(0, \sigma)$, Algorithm 1 is $\varepsilon$-DP with $\varepsilon = T \cdot \log(1 + p \cdot (e^{1/\sigma} - 1))$.*

*Proof Overview.* We first provide the privacy analysis for each step. At each step, line 10 upper bounds the $\ell_1$ (and therefore $\ell_p \forall p \geq 1$ for the scalar value) sensitivity of the difference in losses $l_i$ by $C$. Lines 13 and 15 add noise based on DP mechanisms such that each step in Algorithm 1 satisfies $(\varepsilon, \delta)$-DP with some privacy parameters. For Laplace mechanism, each step satisfies $\varepsilon$-DP where $\varepsilon = \log(1 + p \cdot (e^{1/\sigma} - 1))$. For Gaussian mechanism, each step satisfies $(\varepsilon, \delta)$-DP where $\max(H_{e^\varepsilon}(\mathcal{N}(0, \sigma^2) \| (1-p)\mathcal{N}(0, \sigma^2) + p\mathcal{N}(1, \sigma^2)), H_{e^\varepsilon}((1-p)\mathcal{N}(1, \sigma^2) + p\mathcal{N}(0, \sigma^2) \| \mathcal{N}(1, \sigma^2))) \leq \delta$ and $H_{e^\varepsilon}(\cdot \| \cdot)$ is Hockey-stick divergence (see Definition A.3 in Appendix A). Therefore Algorithm 1 is $(\varepsilon, \delta)$-DP with some privacy parameters that are calculated via some mechanism-dependent composition theorem. Most of the privacy analyses in Appendix A are similar to the privacy analysis of DP-SGD and are based on the numerical composition of privacy loss random variable (see Definition A.7 in Appendix A) for the corresponding privacy curve (Gopi et al., 2021) except for the pure $\varepsilon$-DP analysis. The analysis for pure $\varepsilon$-DP by the Laplace mechanism is based on the basic composition (Dwork & Roth, 2014). Given a number of iterations $T$ and the use of the Laplace mechanism $Laplace(0, C\sigma)$, Algorithm 1 is $\varepsilon$-DP with $\varepsilon = T \cdot \log(1 + p \cdot (e^{1/\sigma} - 1))$.

*Remark.* We can get a tighter composition of $\varepsilon$ by relaxing Laplace mechanism's $\varepsilon$-DP to $(\varepsilon, \delta)$-DP. Note that since we are dealing with scalar values, our mechanism in each iteration will be a one dimensional Laplace mechanism. Therefore we can compute the dominating pair for a single dimensional Laplace mechanism based on Zhu et al. (2022); Wang et al. (2023), that is tighter than directly using the private random variable for privacy curve of a $\varepsilon$-DP algorithm in Gopi et al. (2021). We detail the full privacy analysis in Appendix A.

Prior works (Song et al., 2021; Li et al., 2022a) have analyzed when the convergence rate of DP-SGD is dependent on the effective rank $r$ of the problem rather than the model dimension size $d$. A concurrent and independent work (Zhang et al., 2024a) propose DPZero by privatizing the loss difference scalar in ZO with Gaussian noise, that shares some similar design as our DP-ZO. They provide the convergence guarantee for DPZero that is independent of the model dimension in private training. Here we briefly discuss the algorithm design difference between DPZero in Zhang et al. (2024a) and our DP-ZO. DPZero is Algorithm 2 in Zhang et al. (2024a), and the random perturbation is by sampling $\vec{z}$ from a unit Sphere $\mathbb{S}^{d-1} = \{x \in \mathbb{R}^d | \|x\| = 1\}$. In our Algorithm 1, $\vec{z}$ is sampled from Gaussian distribution. We reimplemented our perturbation method based on the algorithms in Zhang et al. (2024a), and we obtain $82.32_{0.82}$ for LoRA fine-tuning OPT-13B on SQuAD with $(1, 10^{-5})$-DP, that is comparable as the perturbation method in our Algorithm 1, that achieves $82.28_{0.84}$ under the same setting. We also note that Zhang et al. (2024a) also share same observation on this by experiments on Roberta-large models (Liu et al., 2019) in their Table 7. Algorithm 2 in Zhang et al. (2024a) for DPZero is under full batch setting.[2] As we show in Appendix D.2, in the evaluated number of iteration range, given the same amount of computation time on a single GPU, by training more steps with small batch size, DP-ZO achieves better performance than by accumulating steps for large batch size. This result is

---

[2]The experiment section in Zhang et al. (2024a) also uses small batch size and perturbation from random Gaussian distribution.

consistent with the observation in the non-private zeroth-order optimization results in MeZO (Malladi et al., 2023). Besides, Algorithm 2 in DPZero considers the sensitivity as $2C$ with clipping threshold $C$, this leads to adding twice necessary noise under add/remove DP. Our Algorithm 1 analyzes the sensitivity $C$ with clipping threshold $C$, and we share this add/remove DP choice and sensitivity $C$ with popular DP-SGD library like Opacus (Yousefpour et al., 2021). We further discuss our work and Zhang et al. (2024a) in Section 5.

# 4 EVALUATION

We first overview our experimental setup in Section 4.1 and then evaluate the performance of DP-ZO in Section 4.2. We find that DP-ZO provides a competitive privacy-utility trade-off for conservative privacy budgets across multiple datasets, model architectures and can scale to large models under conservative privacy budgets. We also compare DP-ZO to DP-SGD in Section 4.2 and show that DP-ZO achieves comparable performance to DP-SGD for the same model size. Furthermore, we show that DP-ZO achieves a non-trivial privacy-utility trade-off under pure $\varepsilon$-DP under a conservative privacy budget like $\varepsilon = 4$ on large language models. In Section 4.3 we first provides results of DP-ZO across different model architectures. We then measure the empirical privacy loss and computation efficiency of DP-ZO. We also characterize DP-ZO under different few-shot settings and different noise mechanisms for $(\varepsilon, \delta)$-DP.

## 4.1 Experimental Setup

We report the metric of interest (F1 score or accuracy) and standard deviation averaged across 5 independent runs with different random seeds. We detail the full hyper-parameter searches and computation cost in Appendix D.2.

**Datasets.** Following Malladi et al. (2023), we mainly consider three different benchmark NLP tasks: SQuAD (Rajpurkar et al., 2016) and DROP (Dua et al., 2019) for text generation, and SST2 (Socher et al., 2013) for text classification. We use F1 for text generation and accuracy for text classification as evaluation metric (we include a detailed description of the metric in Appendix D.1). Although all these datasets have very different dataset sizes, we consider the *few-shot* setting for all these datasets where we sample 1000 points for each dataset. Fine-tuning LLMs with $O(n = 1000)$ samples is a standard setting in the NLP community (Gao et al., 2021; Malladi et al., 2023) because we are generally interested in the few-shot abilities of LLMs (Brown et al., 2020). This represents a departure from prior works that privately finetune LLMs; Yu et al. (2022); Li et al. (2022b); Yu et al. (2021b) use the entire training dataset of SST2 that has about 65,000 examples. It is well known that the privacy-utility trade-off improves greatly with more data (Tramèr & Boneh, 2021). It is straightforward to see that our setting with datasets of the size $n = 1000$ with $\delta = 10^{-5}$ is simultaneously more challenging and more aligned with real-world usecases than previous works in DP finetuning of LLMs. *Despite the increased difficulty of our few-shot setting as compared to prior work, our results validate that DP-ZO realizes a strong privacy-utility trade-off.* We also varies the training sample size from the few-shot to the full training set by conducting experiments on the QNLI (Wang et al., 2019) dataset to be consistent with previous works (Li et al., 2022b; Yu et al., 2022) for a fair comparison.

**Models.** We present our main results (Table 1) using a pretrained OPT-13B (Zhang et al., 2022) model that is finetuned with LoRA (Hu et al., 2022); that is, we update $< 1\%$ of the total parameters. We include a range of analysis, including varying the model size among the OPT series, model architectures including Mistral-7B-v1 (Jiang et al., 2023) and amount of parameters to be updated, after we present the main results. We also include one experiments for QNLI on RoBERTa-base (Liu et al., 2019) to be consistent with previous works (Li et al., 2022b; Yu et al., 2022) for a fair comparison.

**Privacy budgets.** We consider various privacy levels with $\varepsilon = [0.5, 1, 4]$ and fix $\delta = 10^{-5}$ for $(\varepsilon, \delta)$-DP. We include the zero-shot $\varepsilon = 0$ that does not incur any privacy loss because we evaluate the pretrained model directly without finetuning on private data. We also include the non-private $\varepsilon = \infty$ baseline that is trained without any DP guarantee. That is, we iterate over the shuffled dataset instead of doing Poisson sampling (replacing line 3), do not clip the step size (skipping line 10) and set $\sigma = 0$ (in line 13). We make these modifications because Poisson sampling and small threshold like $C = 0.1$ for per-example clipping makes

gradient estimator biased and loss convergence issues in $\sigma = 0$ for DP-SGD (Andrew et al., 2021; Chen et al., 2020; De et al., 2022; Bu et al., 2023b), and we want to compare to the strongest possible nonprivate baseline.

## 4.2 Main Results

**DP-ZO provides a strong privacy-utility trade-off for conservative privacy budgets.** As shown in Table 1, across all three tasks and all $\varepsilon$s, DP-ZO significantly improves upon the $\varepsilon = 0$ baseline, and only slightly degrades the performance compared to the non-private baseline. For SQuAD, even at $\varepsilon = 0.5$, DP-ZO can still achieve 80.10%, that significantly outperforms $\varepsilon = 0$ baseline (46.23%). The gap between $\varepsilon = 0.5$ and $\varepsilon = \infty$ is about 6.75%. By increasing $\varepsilon$ from 0.5 to 4, this gap can be further reduced to 3%. For DROP and SST2, DP-ZO (Gaussian) achieves comparable performance as the non-private baseline at $\varepsilon = 4$.

Table 1: Main results with 1000 training samples for each dataset. OPT-13B model with LoRA fine-tuning. DP-ZO (G) is DP-ZO instantiated with the Gaussian mechanism. $\delta = 10^{-5}$. The $\varepsilon = \infty$ by ZO is 86.85 for SQuAD, 33.22 for DROP, and 93.69 for SST2. The $\varepsilon = 0$ baseline, i.e., directly doing model evaluation without training, is 46.23 for SQuAD, 14.64 for DROP, and 58.83 for SST2. The results of DP-SGD on DROP are omitted because fine-tuning OPT-13B on the DROP dataset by LoRA will cause the out of memory issue on a single A100 GPU even in the non-private setting.

| Task | SQuAD | | | DROP | | | SST2 | | |
|---|---|---|---|---|---|---|---|---|---|
| Task type | generation (metric: F1) | | | | | | classification (metric: accuracy) | | |
| Method | $\varepsilon = 0.5$ | $\varepsilon = 1$ | $\varepsilon = 4$ | $\varepsilon = 0.5$ | $\varepsilon = 1$ | $\varepsilon = 4$ | $\varepsilon = 0.5$ | $\varepsilon = 1$ | $\varepsilon = 4$ |
| DP-ZO(G) | $80.10_{0.63}$ | $82.28_{0.84}$ | $83.87_{0.50}$ | $28.39_{0.82}$ | $30.30_{0.51}$ | $31.99_{0.51}$ | $85.41_{2.91}$ | $91.19_{0.90}$ | $92.59_{0.30}$ |
| DP-SGD | $79.85_{0.89}$ | $82.14_{0.18}$ | $83.05_{0.51}$ | − | − | − | $64.33_{6.47}$ | $90.25_{0.78}$ | $92.06_{0.52}$ |

**DP-ZO scales to large models.** In Table 2 we show that DP-ZO continues improving as the model size increases from 1.3B to 66B. Due to space constraints, we provide the non-private ($\varepsilon = \infty$) performance of all models and methods in Appendix E. Table 2 shows an promising insight: *as the model size and non-private performance increase, the gap in performance between private and non-private models shrinks.* Specifically, the gap for OPT-1.3B is 5.68% (80.97% at $\varepsilon = \infty$ reduced to 75.29% under $\varepsilon = 1$). But this gap shrinks to just 3.37% for OPT-66B, where the private performance at $\epsilon = 1$ is 84.12% compared to 87.49% non-privately. Our findings suggest that DP-ZO scales to large models not only because it is compatible with existing pipeline without much additional engineering effort but also because the utility drop due to privacy is smaller as the model size increases.

Table 2: DP-ZO (Gaussian) and DP-SGD with full parameter and LoRA fine-tuning on SQuAD with 1000 training samples across different model sizes. $(1, 10^{-5})$-DP. '−' means the approach did not scale with straightforward implementation; Section 5 details the additional engineering required to scale DP-SGD to larger models. '−−' for DP-ZO means the results are omitted due to limited computational resources. Due to limited computing resources, this table does not include the standard deviation for OPT-66B model.

| Method | OPT-1.3B | OPT-2.7B | OPT-6.7B | OPT-13B | OPT-30B | OPT-66B |
|---|---|---|---|---|---|---|
| DP-ZO-LoRA (Gaussian) | $75.29_{0.90}$ | $80.34_{1.14}$ | $81.34_{1.04}$ | $82.28_{0.84}$ | $82.48_{0.83}$ | $84.12_{1.01}$ |
| DP-SGD-LoRA | $75.39_{0.33}$ | $79.42_{0.57}$ | $79.53_{0.52}$ | $82.14_{0.18}$ | − | − |
| DP-ZO-Full (Gaussian) | $72.84_{1.03}$ | $77.25_{0.27}$ | $79.06_{0.67}$ | $82.16_{0.41}$ | −− | −− |
| DP-SGD-Full | $75.50_{0.89}$ | $79.81_{0.64}$ | − | − | − | − |

**Comparison with DP-SGD.** We compare DP-ZO to differentially private stochastic gradient descent (DP-SGD) (Abadi et al., 2016) which has been applied to fine-tune LLMs with full parameter fine-tuning (Li et al., 2022b) and with LoRA (Yu et al., 2022; He et al., 2023). Recall that DP-ZO is compatible out-of-the-box with mixed precision training and GPU parallelism, enabling us to fine-tune OPT-66B. As we discuss in Section 5, it is significantly more challenging to integrate DP-SGD with these techniques, and furthermore, DP-SGD

requires more memory than DP-ZO to store activations and compute per-sample gradients (see Section 4.3). As a direct result, DP-SGD cannot directly scale past 2.7B with full fine-tuning or 13B with LoRA without additional implementation effort for multi-GPU training, while DP-ZO can scale seamlessly to larger models. In Table 2 we present comparisons between DP-ZO and DP-SGD with full parameter finetuning and LoRA. With the same model size, DP-ZO achieves comparable performance as DP-SGD as by LoRA finetuning, i.e., both DP-ZO and DP-SGD achieves 82% on OPT-13B models. The best performance by DP-ZO is 84.12% by OPT-66B finetuned with LoRA. This is $\approx 2\%$ better than the best performance of DP-SGD in Table 2 that is 82.14% by OPT-13B with LoRA.

**DP-ZO with pure $\varepsilon$-DP.** To the best of our knowledge, DP-ZO (Laplace) is the first method that achieves a non-trivial privacy-utility trade-off under pure $\varepsilon$-DP under a conservative privacy budget like $\varepsilon = 4$ on large language models.

In Table 3, DP-ZO (Laplace) can significantly improve upon $\varepsilon = 0$. Given a budget $\varepsilon = 4$, which some prior work has considered reasonable (Ponomareva et al., 2023), DP-ZO (Laplace) can obtain 73.52% on SQuAD. When increasing $\varepsilon = 4$ to $\varepsilon = 15$, DP-ZO (Laplace) can obtain 78.82% on SQuAD. Note that the $l_1$ sensitivity required for Laplace mechanism makes it hard for DP-SGD to achieve comparable performance as DP-ZO because the gradients in DP-SGD have high dimension. Table 3 shows that DP-SGD with $l_1$ norm clipping and Laplace noise only achieves 47.25% for reasonable privacy budgets with $\varepsilon$ ranging from 4 to 15, that is only marginal improvement upon the zero-shot performance. Even when relaxing the privacy budget to near-vacuous guarantees such as $\varepsilon = 10450$, DP-SGD (Laplace) still achieves worse performance compared to DP-ZO due to the Laplace noise added in high-dimension gradients.

Table 3: Pure $\varepsilon$-DP by DP-ZO (Laplace), SQuAD with 1000 training samples. OPT-13B with LoRA fine-tuning. The $\varepsilon = 0$ baseline is 46.23%.

| $\varepsilon$ | $\varepsilon = 4$ | $\varepsilon = 10$ | $\varepsilon = 15$ | $\varepsilon = 10450$ |
|---|---|---|---|---|
| DP-ZO (Laplace) | $73.52_{1.04}$ | $76.75_{1.39}$ | $78.82_{1.57}$ | $81.02_{1.24}$ |
| DP-SGD (Laplace) | $47.25_{0.79}$ | $47.27_{0.95}$ | $47.36_{1.02}$ | $76.50_{0.89}$ |

## 4.3 Analysis

In this section, we provide experiments for DP-ZO across different model architectures, empirical privacy analysis to measure how private is DP-ZO, memory efficiency of DP-ZO, the amount of training data that we sample in DP-ZO, and the choice of DP mechanism in DP-ZO.

**DP-ZO provides a strong privacy-utility trade-off across different model architectures.** Table 1 and Table 2 show that DP-ZO achieves the comparable performance as DP-SGD on various OPT models sizes. We now run experiments on SQuAD with Mistral-7B-v1 model (Jiang et al., 2023) in Table 4 and include the results of OPT-6.7B and OPT-13B for the ease of comparison. On Mistral-7B-v1, DP-ZO and DP-SGD both achieve comparable performance at $\varepsilon = 1$, that are around 89%. Moreover, even Mistral-7B-v1 has similar model parameters as OPT-6.7B and much fewer parameters than OPT-13B, DP-ZO achieves better performance by Mistral-7B-v1 than OPT-13B. This indicates that with the development of more advanced models, the power of DP-ZO will be further unlocked.

Table 4: The results of DP-ZO on different model architectures. The $\varepsilon = 0$ baseline for Mistral-7B-v1 is 68.37.

| Models | OPT-6.7B | OPT-13B | Mistral-7B-v1 |
|---|---|---|---|
| DP-ZO | $81.34_{1.04}$ | $82.28_{0.84}$ | $89.79_{0.41}$ |
| DP-SGD | $79.53_{0.52}$ | $82.14_{0.18}$ | $89.44_{0.41}$ |

**DP-ZO is memory efficient.** We omitted several results of DP-SGD in Section 4.2 due to excessive memory consumption of DP-SGD that leads to out-of-memory (OOM) issue on a single A100 80G GPU. We now provide a more fine-grained memory analysis for a better understanding of DP-ZO and DP-SGD.

The naive implementation of DP-SGD causes additional memory consumption due to the per-example gradient computation. An ongoing line of work (Li et al., 2022b; Yu et al., 2022; He et al., 2023; Bu et al., 2024) has continued improving the scalability of DP-SGD over the past few years. For memory cost comparison, we consider several variants of DP-SGD including DP-SGD-full (Abadi et al., 2016; Li et al., 2022b), DP-SGD-full(ghost) (Li et al., 2022b), DP-SGD-LoRA (Yu et al., 2022), DP-SGD-BitFiT (Bu et al., 2024) and DP-ZO (including full and LoRA) for a fair comparison.[3]

As discussed in Li et al. (2022b); De et al. (2022), small batch size will incur sub-optimal performance of DP-SGD and therefore large batch size is preferred, we consider gradient accumulation in memory analysis to enable large batch size. We consider full precision both for DP-SGD and DP-ZO for fair comparison. We present the results of such memory consumption for different sequence lengths on OPT-2.7B in Table 5.

Table 5: Comparison of memory consumption (GB) of DP-ZO and DP-SGD varying different sequence length $L$ on OPT-2.7B. Full-precision, no gradient check-pointing. Batch size=2, gradient accumulation steps=2. OOM indicates out-of-memory on a single A100 80G GPU.

| Methods | DP-SGD-full | DP-SGD-full(ghost) | DP-SGD-LoRA | DP-SGD-BitFit | DP-ZO-full | DP-ZO-LoRA |
|---|---|---|---|---|---|---|
| $L = 128$ | 51.3 | 32.9 | 11.4 | 12.2 | 11.6 | 10.3 |
| $L = 512$ | 51.4 | 39.6 | 18.1 | 21.3 | 11.7 | 11.1 |
| $L = 2048$ | OOM | OOM | OOM | OOM | 15.6 | 15.6 |

Table 5 shows that the naive implementation of DP-SGD incurs around 50GB for sequence length $L = 128$ and incurs out-of-memory (OOM) issue when increasing sequence length to $L = 2048$. Ghost clipping can help reduce the memory consumption by removing per-sample gradient computation, but still needs to accumulate gradients and consumes memory more than 30GB. Parameter efficient fine-tuning methods including DP-SGD-LoRA, DP-SGD-BitFiT can largely reduce the memory cost for gradient accumulation with fewer parameters in gradients. However, as the input sequence length $L$ increases, the activation saved in forward-pass for gradients computation significantly increases and inputs with sequence length $L = 2048$ still cause OOM error for parameter efficient fine-tuning methods. Note that such long sequence input exists in practical scenarios such as digesting from a long documents and recent foundation models put efforts to support longer context (Gemini-Team et al., 2023). In contrast, DP-ZO does not need to store gradients nor store additional activation for gradients, therefore can reduce the memory consumption to be less than 16GB even when sequence length is 2048. In fact, DP-ZO only incurs a negligible additional memory cost than ZO (Malladi et al., 2023) for the per-sample loss clipping and noise addition.

We now consider a more restrictive memory setting such as on-device mobile setting, i.e., 8GB as a memory limit (Gim & Ko, 2022; Guo et al., 2024). As discussed in Malladi et al. (2023), gradient checkpointing can help reduce the activation memory consumption. We use gradient checkpointing to reduce the memory consumption of activation and half-precision to reduce the memory consumption of weights and gradients. We measure the memory cost with the default implementation of activation checkpointing where we checkpoint every block of the model. We consider batch size=1 and accumulate gradients in 2 steps to enable training in large batch size. We present the results of the memory consumption for different sequence lengths on OPT-2.7B model in Figure 5. While gradient checkpointing significantly reduces the memory consumption in DP-SGD, DP-SGD-full is still beyond the 8GB memory limit due to gradient accumulation. Similarly, with gradient checkpointing, DP-SGD-

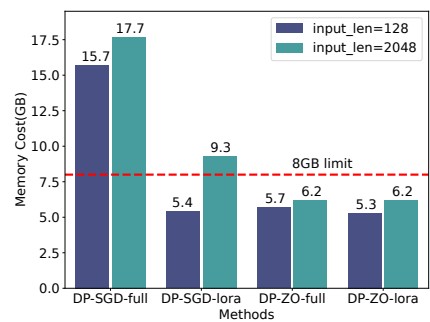

Figure 5: Memory comparison of DP-ZO and DP-SGD with half-precision and gradient checkpointing. Batch size=1, gradient accumulation steps=2.

---

[3]DP-SGD-full, DP-SGD-full(ghost), DP-SGD-LoRA are based on https://github.com/lxuechen/private-transformers. For DP-SGD-BitFit, we use fastDP https://github.com/awslabs/fast-differential-privacy/tree/main/fastDP and follow the guideline by using one line of code [param.requires_grad_(False) for name, param in model.named_parameters() if '.bias' not in name].

LoRA still exceeds the 8GB limit when increasing sequence length to 2048. In contrast, DP-ZO incurs just 6.2GB even when input sequence length is 2048.

**How private is DP-ZO and ZO: an empirical privacy analysis.** We discussed earlier in Section 2.2 that the single scalar information from Zeroth-order Optimization will leak private information and therefore motivate our design of DP-ZO. We now validate our design by conducting empirical privacy evaluation through membership inference attacks (MIA) (Shokri et al., 2017) to understand the privacy implication of DP-ZO. We use the state-of-the-art privacy auditing method in Panda et al. (2024b) for empirical privacy evaluation. Similar to Panda et al. (2024b), we construct synthetic canaries by creating one new token for each canary to the vocabulary to increase the information from canaries for better auditing. Following Panda et al. (2024b); Mireshghallah et al. (2022), we use MIA Area under the ROC Curve (AUC-ROC) for empirical privacy evaluation. We report the results for DP-ZO and DP-SGD in Table 6 for different $\varepsilon$s including $[0.5, 1, 4, 10, \infty]$.

As observed in Panda et al. (2024b), this privacy attack method is very successful for DP-SGD when no noise added, and DP-SGD can effectively reduce the attack AUC to 54.8 that is closed to random guess at $\varepsilon = 0.5$. For DP-ZO, when no noise added, the MIA AUC is 71.4 that is lower than DP-SGD($\varepsilon = \infty$), however much higher than the random guess baseline 50. Interestingly, in this empirical privacy case study, the privacy leakage of ZO is similar to the privacy leakage of DP-SGD at $\varepsilon = 10$. DP-ZO is motivated to get a formal privacy guarantee by differential privacy. DP-ZO can reduce the attack AUC to around random guess at $\varepsilon = [0.5, 1, 4]$. To the best of our knowledge, this is the first experimental result that measures the privacy leakage in ZO and DP-ZO. This result shows the necessity of DP-ZO to reduce MIA close to random guess, and also raises an open problem about the inherent privacy property of zeroth-order optimization.

Table 6: Membership Inference Attack AUC-ROC for DP-ZO and DP-SGD.

|  | $\varepsilon = 0.5$ | $\varepsilon = 1$ | $\varepsilon = 4$ | $\varepsilon = 10$ | $\varepsilon = \infty$ |
|---|---|---|---|---|---|
| DP-ZO | 53.2 | 54.5 | 55.6 | 58.8 | 71.4 |
| DP-SGD | 54.8 | 55.0 | 61.5 | 71.9 | 100.0 |

**Characterizing the effect of data size.** Although it is known that private learning requires more data than non-private learning (Bassily et al., 2014), prior work has not characterized this improvement for fine-tuning language models. In Table 7 and Figure 6 we first vary the number of training samples $n$ around the $n = 1000$ setting in the main results while keeping $\delta = 10^{-5}$ fixed for all choices of $n$. Table 7 shows that DP-ZO can achieve nontrivial performance in few-shot settings under conservative privacy guarantees. Furthermore, we find that while increasing the amount of training data by $10\times$ barely increases non-private performance, it increases private performance by $\approx 6\%$ ($n = 500$ vs. $n = 5000$). Similarly, for $\varepsilon$-DP, acquiring more data enhances privacy amplification and reduces the amount of noise we need to add to achieve a target DP $\varepsilon$ guarantee. In Table 8 we find that increasing the number of training examples from 1000 to 5000 improves performance at $\varepsilon = 4$ from 73.52% to 79.89%, although the improvement of

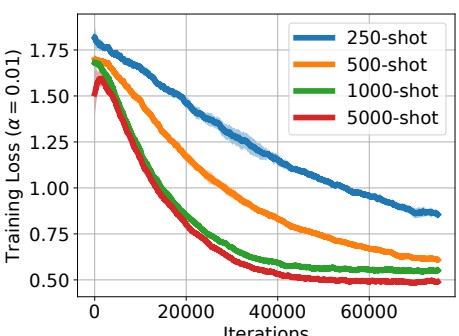

Figure 6: (Smoothed) training loss. $n = 5000$ has better convergence rate compared to $n = 250$.

non-private performance at $\epsilon = \infty$ by increasing training samples from 1000 to 5000 is insignificant.

While non-private few-shot learning can succeed by just memorizing the training data, Figure 6 indicates that the convergence rate for different shots for private few-shot learning is different. With the proliferation of pretrained models, we anticipate that *privately fine-tuning downstream tasks in the few-shot setting will be more aligned with real-world use cases (Gao et al., 2021).*

Besides the few-shot setting, we now scale the size of training data size up to more than 100k samples and compare the performance of DP-SGD and DP-ZO. For the ease of comparison, we follow Li et al. (2022b); Yu

Table 7: The effect of different $n$ training samples for DP-ZO (Gaussian) fon SQuAD dataset. $(1, 10^{-5})$-DP. OPT-13B with LoRA finetuning.

| $n$-shot | $n = 250$ | $n = 500$ | $n = 1000$ | $n = 5000$ |
|---|---|---|---|---|
| $\varepsilon = 1$ | $74.86_{0.74}$ | $78.25_{2.38}$ | $82.28_{0.84}$ | $84.29_{0.92}$ |
| $\varepsilon = \infty$ | $86.40$ | $86.53$ | $86.85$ | $86.92$ |

Table 8: $\varepsilon$-DP ($\varepsilon = 4$) by DP-ZO (Laplace) on different $n$ in SQuAD on OPT-13B (LoRA). . $\varepsilon = \infty$ by ZO is 86.85% for $n = 1000$ and 86.92% for $n = 5000$. $\varepsilon = 0$ is 46.2%.

| $n$-shot | $n = 1000$ | $n = 5000$ |
|---|---|---|
| DP-ZO (Laplace) | $73.52_{1.04}$ | $79.89_{0.49}$ |

Table 9: Comparison of DP-ZO and DP-SGD on different $n$ in QNLI on RoBERTa-base model. $(3, 4.7 \times 10^{-6})$-DP.

| $n$-shot | 1000 | 5000 | 10000 | 50000 | 104743 |
|---|---|---|---|---|---|
| DP-SGD | 73.27 | 79.44 | 80.54 | 84.81 | 87.40 |
| DP-ZO | 76.19 | 78.66 | 79.28 | 79.70 | 79.85 |

Table 10: DP-ZO with different DP mechanism. SQuAD with 1000 training samples. $\delta = 10^{-5}$.

| $\varepsilon$ | $\varepsilon = 0.5$ | $\varepsilon = 1$ | $\varepsilon = 4$ |
|---|---|---|---|
| DP-ZO (G) | $80.10_{0.63}$ | $82.28_{0.84}$ | $83.87_{0.50}$ |
| DP-ZO (L) | $77.58_{0.81}$ | $80.49_{0.63}$ | $82.94_{0.69}$ |

et al. (2022) and conduct experiments for DP-ZO and DP-SGD on QNLI with RoBERTa-base with a range of examples at $(3, 4.7 \times 10^{-6})$-DP and report result in Table 9. Similar to previous observation, Table 9 shows that at a small data regime, i.e., number of samples $n = 1000$, DP-ZO achieves comparable performance as DP-SGD. DP-ZO can also improve its performance within more samples. Compared to DP-SGD, we notice that when increasing the data size $n$, there is a utility gap between DP-ZO and DP-SGD. In addition, we compare the utility of DP-ZO, ZO, DP-SGD, SGD under different $n$ in Table 11 to further understand the utility gap between DP-ZO and DP-SGD. Due to the limited computation resources, we provide results for RoBERTa-base on QNLI and OPT-13B (LoRA fine-tuning) on SST2 in Table 11 for $n = 1000$ and full training set. We observe a similar utility gap between DP-ZO and DP-SGD as QNLI in Table 9 for SST2 in Table 11 when increasing $n$ to the full training set. The utility gap of SGD and ZO on QNLI and SST2 in Table 11 indicate that the challenge of utility for ZO when the number of training samples $n$ increases. Therefore there is also the utility gap between DP-ZO and DP-SGD when $n$ increases. This limitation shows an open problem on how to improve data efficiency in zeroth-order optimization (Zhang et al., 2024c; Zhao et al., 2024) and DP-ZO, and we leave this data efficiency improvement in ZO as future work.

**Different noise mechanisms for $(\varepsilon, \delta)$-DP.** We now relax the privacy guarantee provided by the Laplace mechanism to approximate $(\varepsilon, \delta)$-DP. In Table 10, we compare DP-ZO instantiated with the Laplace and Gaussian mechanisms. DP-ZO (Gaussian) outperforms DP-ZO (Laplace) for strict privacy budgets such as $\varepsilon = 0.5$ because it enjoys tighter accounting (Gopi et al., 2021) and lower variance (Dwork & Roth, 2014). These advantages are less significant for larger privacy budgets; for $\varepsilon = 4$, the gap between DP-ZO (Gaussian) and DP-ZO (Laplace) is within 1%.

Our experiments on the DP-ZO with laplace for $\varepsilon$-DP and the comparisons of Laplace and Gaussian mechanisms for $(\varepsilon, \delta)$-DP shows that DP-ZO provides a strong privacy-utility trade-off under different DP mechanisms while DP-SGD suffers from Laplace mechanisms for $(\varepsilon, \delta)$-DP, which opens the new opportunity for the synergy between DP mechanisms and large language models.

Table 11: (DP-)ZO and (DP-)SGD across different number of training samples $n$. $(3, 4.7 \times 10^{-6})$ for QNLI on RoBERTa-base. $(1, 10^{-5})$ for SST2 on OPT-13B. For full set, $n = 104743$ for QNLI and $n = 66849$ for SST2.

| | QNLI | | | | SST2 | | | |
|---|---|---|---|---|---|---|---|---|
| | DP-ZO | ZO | DP-SGD | SGD | DP-ZO | ZO | DP-SGD | SGD |
| $n = 1000$ | 76.19 | 79.85 | 73.27 | 84.40 | 91.19 | 93.69 | 90.25 | 95.64 |
| full set | 79.85 | 79.99 | 87.40 | 89.49 | 93.00 | 94.84 | 95.07 | 96.33 |

# 5 DISCUSSION

**Discussion of our work and DPZero (Zhang et al., 2024a).** Most recently, a concurrent work (Zhang et al., 2024a) also privatizes the SPSA algorithm in zeroth-order optimization method. Zhang et al. (2024a) propose DPZero and provide the convergence guarantee for DPZero that is independent of the model dimension in private training. Zhang et al. (2024a) presented experimental results for DPZero/DP-SGD on Roberta-large model and DPZero on OPT 1.3B-6.7B models. Our work focus on understanding the privacy-utility trade-off for DP-ZO as well as the privacy implication and the practical implication of DP-ZO. We provide a comprehensive understanding of the privacy-utility trade-off for DP-ZO and DP-SGD for OPT-1.3B to 66B models. We consider DP-ZO as a general framework for private machine learning. We found that DP-ZO is compatible with Laplace mechanism and enabling $\varepsilon$-DP with a reasonable privacy-utility trade-off while DP-SGD with Laplace mechanism suffers to optimize. The result indicates that DP-ZO can open the new opportunity for the synergy between DP mechanisms and large language models. To the best of our knowledge, we provide the first experimental result that measures the privacy leakage in (DP-)ZO. We also conduct experiments for analyzing the effect of data size and the memory consumption for limited resource scenarios. Our findings could provide insights on future directions for improving the analysis and implementation of DP-ZO methods.

**Discussion on DP-SGD and DP-ZO.** In Section 4.2 we showed that DP-ZO obtains competitive privacy-utility tradeoff. Now we examine the amount of engineering effort necessary to scale DP-SGD to larger models, a topic on which many papers have been written (Bu et al., 2023b;c; Yousefpour et al., 2021; Li et al., 2022b; He et al., 2023; Bu et al., 2023a). We find that DP-ZO *seamlessly scales to larger models* and believe its simplicity presents a compelling alternative to DP-SGD for practitioners.

**DP-SGD.** Differentially Private Stochastic Gradient Descent (DP-SGD) (Song et al., 2013; Abadi et al., 2016) is the standard privacy-preserving algorithm to train models on private data, with an update rule given by $w^{(t+1)} = w^{(t)} - \frac{\eta_t}{|B|} \left( \sum_{i \in B} \frac{1}{c} \texttt{clip}_c(\nabla \ell(x_i, w^{(t)})) + \sigma \xi \right)$ where the two changes to SGD are the per-sample gradient clipping $\texttt{clip}_c(\nabla \ell(x_i, w^{(t)})) = \frac{c \times \nabla \ell(x_i, w^{(t)})}{\max(c, ||\nabla \ell(x_i, w^{(t)})||_2)}$ and addition of noise sampled from a $d$-dimensional Gaussian distribution $\xi \sim \mathcal{N}(0, 1)$ with standard deviation $\sigma$. DP-SGD is the marquee algorithm for privacy-preserving machine learning, but it requires implementing per-example gradient clipping. This creates a slew of challenges for deploying DP-SGD.

**Computational and memory challenges in DP-SGD.** DP-SGD requires the computation of per-example gradients, which can be naively implemented by storing each gradient in the batch separately. This approach inflates the memory overhead by a factor of $B$, where $B$ is the batch size. TensorFlow Privacy avoids this issue by clipping microbatches rather minibatches, which does not slow down training but increases the noise added and therefore hurts utility. Jax can automatically vectorize the per-sample gradient computation, but training is still slowed down. Recently, specialized libraries have been developed that instead analytically compute the norm of the gradients for different layers (Li et al., 2022b; Bu et al., 2023d; Ding et al., 2024). This requires actually implementing the computation, which is challenging for new layers. Parameter efficient fine-tuning methods (Yu et al., 2022; Bu et al., 2024) can help reduce the computation cost by reducing the number of trainable parameters. However, as discussed in Section 4.3, those methods still incur much more memory cost than DP-ZO for long sequences. Besides, when model cannot be loaded into a single GPU, model parallelism is needed to load large models across several GPUs and per-example gradient norm clipping requires additional implementation, both for gradient clipping and communication across device. He et al. (2023) investigate how to make group-wise gradient clipping efficient and achieve good performance in DP-SGD and fine-tunes 175B GPT3 model with 16 V100 GPUs each with 32 gigabytes of VRAM. Bu et al. (2023a) implements DP-SGD based on Zero Redundancy Optimizer (Rajbhandari et al., 2020) to scale model size up to GPT-100B and maintain efficiency. It currently also only supports layer-wise or block-wise clipping.

We now discuss the advantage of DP-ZO that can scale to large language models seamlessly. Note that DP-ZO inherits the seamless scalability from ZO as a result of only additional computation cost on loss. DP-ZO achieves comparable performance as DP-SGD. In contrast, DP-SGD incurs more computational and memory challenges than SGD due to per-example gradient clipping.

**Model parallelism and data parallelism in DP-ZO.** To train large models like OPT-66B, whose parameters cannot be loaded into memory on a single A100 GPU, we need to implement some form of parallelism across GPUs. It is easy for such parallelism in DP-ZO (in the simplest form, just running DP-ZO on a machine with 2 GPUs will prompt HuggingFace to implement naive model parallelism), while much more effort in DP-SGD. To synchronize model state between GPUs in data-parallel-DP-ZO, we just transfer the random seed and its corresponding half-precision float16 scalar step size; this is just a few bytes. However, first-order approaches such as DP-SGD require the transfer of gradients across devices to update all the models, necessitating expensive allgather and reduce operations. This communication overhead is $1.5d$ in PyTorch FSDP, where $d$ is the size of the model.

**DP-ZO is storage and communication-efficient even after training has completed.** DP-ZO offers significant advantages in terms of storage and communication efficiency, especially beneficial for bandwidth-constrained environments like edge devices (Li et al., 2024). Unlike traditional methods where the difference in model parameters $\theta_0 - \theta_f$ is shared which could amount to multiple gigabytes for large models, DP-ZO allows for the storage and transmission of a sequence of updates. This sequence is represented as an array of tuples $[(\text{SEED}_0, 0.54), \cdots, (\text{SEED}_f, -0.14)]$, where each tuple contains a seed and a step size, taking up only 4 bytes. Even for $1 \times 10^4$ fine-tuning iterations, this array would require less than 1MB of storage, representing a substantial reduction in both storage and communication overhead. We can apply these weight differences to a model by simply iterating over the array, sampling from the PRNG using the given seed, scaling that random vector, and applying it to the current model parameters. This procedure is highly efficient, as it involves only sequential memory accesses and scalar floating-point operations.

**DP-SGD vs. DP-ZO.** We have discussed the memory efficiency of DP-ZO over DP-SGD in details. This is helpful in limited memory scenarios as we show in Figure 5. While being much more memory efficient, DP-ZO achieves comparable performance as DP-ZO on several tasks including OPT-13B models on 1000 SQuAD examples with Gaussian mechanism (Table 1) and provides reasonable privacy-utility trade-off for $\varepsilon$-DP (Table 3), which is promising. We also note that the utility challenge of DP-ZO when scaling with more samples (Table 7), mostly due to the gap of ZO and SGD (Table 11). This indicates the future improvement for data efficiency in zeroth-order optimization, that will benefit DP-ZO with Gaussian and Laplace mechanism.

## 6 CONCLUSION

DP-SGD has been the de-facto private training method of the last decade. In this work we propose DP-ZO, a novel method for private fine-tuning that privatizes the zeroth-order update by adding noise to the difference in loss between two perturbations. DP-ZO's unique univariate privatization unlocks training larger models with better parallelism than DP-SGD. DP-ZO provides a strong privacy-utility trade-off across different tasks, model sizes, dataset sizes, and DP mechanisms. We anticipate that future work can further study these design choices, integrate more DP mechanisms into DP-ZO, and apply it to the vision domain.

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

# A    Privacy Analysis

DP-ZO can be instantiated with different noise mechanisms. In this subsection we provide privacy analysis for the Gaussian mechanism and Laplace mechanism.

We first introduce two propositions for DP analysis.

**Proposition A.1** (Basic Composition theorem (Dwork & Roth, 2014))**.** *If $M_1$ is $(\varepsilon_1, \delta_1)$-DP and $M_2$ is $(\varepsilon_2, \delta_2)$, then the adaptive composition of $M_1$ and $M_2$ is $(\varepsilon_1 + \varepsilon_2, \delta_1 + \delta_2)$-DP.*

**Proposition A.2** (Privacy Amplification via Subsampling (Balle et al., 2018))**.** *If $M$ is $(\varepsilon, \delta)$-DP, then the subsampled mechanism with sampling rate $p$ obeys $(\varepsilon', \delta')$-DP with privacy parameters $\varepsilon' = \log(1 + p(e^\varepsilon - 1))$ and $\delta' = p\delta$.*

Analyzing the privacy guarantees of a specific mechanism can be done via worst-case inputs to a mechanism $M$ leading to a pair of worst-case distributions (Dwork & Roth, 2014). Meiser & Mohammadi (2018) introduce a novel method to approximately compose the privacy curves based on the discretized version of privacy loss random variables (Dwork & Rothblum, 2016), whose distribution is called privacy loss distribution (PLD). Meiser & Mohammadi (2018) provide the worst-case pair of distributions for basic mechanisms such as Gaussian mechanism and Laplace mechanism. Sommer et al. (2019) derive the exact analytical and closed formula for the Gaussian mechanism. Zhu et al. (2022) formalize a rigorous notion of the "worst-case" PLD for DP mechanism under the name dominating PLDs. Therefore, following Zhu et al. (2022), we introduce the following definitions.

**Definition A.3** (Hockey-stick Divergence)**.** For $\alpha > 0$, the Hockey-stick divergence is defined as $H_\alpha(P\|Q) := \mathbb{E}_{o \sim Q}[(\frac{P(o)}{Q(o)} - \alpha)_+]$, where $(x)_+ := x\mathbf{1}(x \geq 0)$.

**Lemma A.4.** *(Barthe et al., 2016) For a randomized algorithm $\mathcal{M}$, $\sup_{D \simeq D'} H_{e^\varepsilon}(\mathcal{M}(D)\|\mathcal{M}(D')) \leq \delta$ is equivalent to Definition 2.1.*

We follow Zhu et al. (2022) and formalize $\delta$ as a function of $\varepsilon$.

**Definition A.5** (Optimal Privacy Curve)**.** The *optimal privacy curve* of a mechanism $\mathcal{M}$ is the function $\delta_{\mathcal{M}} : \mathbb{R}^+ \to [0, 1]$ s.t. $\delta_{\mathcal{M}}(\varepsilon) := \sup_{D \simeq D'} H_{e^\varepsilon}(\mathcal{M}(D)\|\mathcal{M}(D'))$.

**Definition A.6** (Dominating Pair of Distributions (Zhu et al., 2022))**.** $(P, Q)$ is a *dominating* pair of distributions for $\mathcal{M}$ (under neighboring relation $\simeq$) if for all $\alpha \geq 0$

$$\sup_{D \simeq D'} H_\alpha(\mathcal{M}(D)\|\mathcal{M}(D')) \leq H_\alpha(P\|Q) \tag{3}$$

When $P, Q$ is chosen such that equation 3 takes "=" for all $\alpha$, $(P, Q)$ is a *tight* dominating pair of distributions or simply, *tightly dominating*.

Based on Sommer et al. (2019); Koskela et al. (2020); Gopi et al. (2021), we follow Zhu et al. (2022) and define the privacy loss random variable as follows.

**Definition A.7** (Privacy Loss Random Variable)**.** We call $Y := \log \frac{P(o)}{Q(o)}, o \sim P$ the *privacy loss random variable* (PRV) for mechanism $\mathcal{M}$ associated with dominating pair $(P, Q)$.

Zhu et al. (2022) prove that any privacy mechanism has a tightly dominating pair of distributions and provide the dominating pairs for basic privacy mechanisms such as Gaussian mechanism and Laplace mechanism.

Further, based on Definitions A.3 and A.7, $H_{e^\varepsilon}(P\|Q)$ can be written as an expectation: $H_{e^\varepsilon}(P\|Q) = \mathbb{E}_Y \left[ \left(1 - e^{\varepsilon - Y}\right)_+ \right]$. $\delta_{\mathcal{M}}(\varepsilon)$ can be bounded by first identifying $\mathcal{M}$'s dominating pair distributions as well as the associated PRV $Y$, and then computing this expectation, denoted as $\delta_Y(\varepsilon) := \mathbb{E}_Y \left[ \left(1 - e^{\varepsilon - Y}\right)_+ \right]$. $\delta_{\mathcal{M}}(\varepsilon) \leq \delta_Y(\varepsilon)$, where $\delta_{\mathcal{M}}(\varepsilon) = \delta_Y(\varepsilon)$ for tight dominating pair $(P, Q)$.

As we sample a batch of samples from the training data via poisson subsampling in each iteration for experiments, we introduce the subsampling amplification for dominating pair following Zhu et al. (2022).

**Proposition A.8.** *(Zhu et al., 2022) Let $\mathcal{M}$ be a randomized algorithm. If $(P, Q)$ dominates $\mathcal{M}$ then $(P, (1-p)P + pQ)$ dominates $\mathcal{M} \circ S_{\text{Poisson}}^p$ for add neighbors and $((1-p)Q + pP, Q)$ dominates $\mathcal{M} \circ S_{\text{Poisson}}^p$ for remove neighbors , where $S_{Poisson}^p$ is poisson sampling with subsampling rate p.*

Given a dominating pair, Zhu et al. (2022) show that if the characteristic function of the PLDs has an analytical expression, for example, Gaussian mechanism, the tight DP composition bounds can be computed in $O(1)$ times. However, without the closed form, such as the sub-sampled Gaussian and sub-sampled Laplace, numerical accounting methods for $T$ steps are proposed to approximate the integral composition formula and compute upper bounds on the privacy budget parameters $(\varepsilon, \delta)$ in DP (Koskela et al., 2020; Koskela & Honkela, 2021; Koskela et al., 2021; Gopi et al., 2021; Doroshenko et al., 2022; Alghamdi et al., 2023; Wang et al., 2023). There are multiple open-source implementations for numerically accurate composition including Microsoft (2021) and Google's-DP-Library (2020) that are used by DP-SGD experiments (Yu et al., 2022; Chua et al., 2024) .

We now provide the privacy analysis for DP-ZO.

**Gaussian Mechanism.** As outlined in Line 10, the $\ell_2$ sensitivity of Algorithm 1 is $C$ and we are adding $\mathcal{N}(0, C^2\sigma^2)$ noise to the estimated loss. From Zhu et al. (2022), we know that the tight dominating pair for Gaussian mechanisms is $P = \mathcal{N}(0, \sigma^2)$ and $Q = \mathcal{N}(1, \sigma^2)$. By Proposition A.8, each step therefore satisfies $(\varepsilon, \delta)$-DP where

$$\max(H_{e^\varepsilon}(\mathcal{N}(0, \sigma^2) \| (1-p)\mathcal{N}(0, \sigma^2) + p\mathcal{N}(1, \sigma^2)), H_{e^\varepsilon}((1-p)\mathcal{N}(1, \sigma^2) + p\mathcal{N}(0, \sigma^2) \| \mathcal{N}(1, \sigma^2))) \leq \delta \quad (4)$$

For $T$ step, we analyze the compositions of subsampled Gaussian for $T$ steps with the PRV accountant of Gopi et al. (2021). We provide the privacy parameters we use for Gaussian mechanism in Table 16 and Table 17. We also provide the example code for using Microsoft PRV Accountant library (Microsoft, 2021) below.

```python
from prv_accountant import PoissonSubsampledGaussianMechanism
from prv_accountant import PRVAccountant
sample_rate = 16/1000
sigma = 16.4
step = 75000
delta = 1/(1e5)
mech = PoissonSubsampledGaussianMechanism(
    noise_multiplier=sigma,
    sampling_probability=sample_rate,)
accountant = PRVAccountant(
    prvs=[mech],
    max_self_compositions=[step+10],
    eps_error=0.001,
    delta_error=1e-10)
eps_low, eps_est, eps_up = accountant.compute_epsilon(delta=delta, num_self_compositions=[
    step])
print(eps_low, eps_est, eps_up)
```

Listing 1: DP accounting using Microsoft PRV Accountant (Microsoft, 2021). We ensure the upper bound $eps_{up}$ does not exceed the desired $\varepsilon$ at the specified $\delta$.

**Laplace Mechanism.** Laplace mechanism can give a pure DP guarantee of $\delta = 0$ which can be of interest in some scenarios. Here we first analyze the pure $\varepsilon$-DP guarantee provided by Laplace mechanism and then provide the analysis for approximate $(\varepsilon, \delta)$-DP analysis.

**Pure $\varepsilon$-DP by Laplace mechanism.** We use data in a single batch instead of all training data to compute the gradients in each updates. For the privacy analysis for Laplace mechanism in Algorithm 1, when we sample each batch in the poisson manner, we could leverage Proposition A.2 to compute the private amplification by subsampling. We first analyze the privacy cost for one step by the Laplace mechanism. At each step, we sample a new batch of data with the sample rate of $p = B/|\mathcal{D}|$. As outlined in Line 10, the $\ell_1$ sensitivity of Algorithm 1 is $C$. By Proposition 2.3, the privacy cost at one step would cost $(1/\sigma, 0)$-DP on this batch. By Proposition A.2, the privacy cost at one step would cost $(\log(1 + p \cdot (e^{1/\sigma} - 1)), 0)$-DP on the full dataset

$\mathcal{D}$. By Proposition A.1, the privacy cost of Algorithm 1 instantiated with Laplace mechanism satisfies $(T \cdot \log(1 + p \cdot (e^{1/\sigma} - 1)), 0)$-DP.

We provide the privacy parameters we used for pure $\varepsilon$-DP by the Laplace mechanism in Table 12.

Table 12: Privacy parameters for Table 3 and Table 8. $\varepsilon$-DP by Laplace. BSZ=20, Steps=2000.

| $\sigma$ | $|D|$ | $\varepsilon$ |
|------|------|----|
| 10.5 | 1000 | 4 |
| 4.5 | 1000 | 10 |
| 3.2 | 1000 | 15 |
| 2.5 | 5000 | 4 |

Table 13: Privacy parameters for Table 10. $(\varepsilon, \delta)$-DP guarantee for Laplace. $\delta = 10^{-5}$. $|\mathcal{D}| = 1000$. BSZ=16, Steps=75000.

| $\sigma$ | $\varepsilon$ (by Monte-Carlo) | $\varepsilon$ (by pure-DP PRV) |
|------|------|------|
| 30.8 | 0.5 | 0.51 |
| 16.3 | 1 | 1.04 |
| 4.6 | 4 | 4.70 |

**Approximate $\varepsilon$-DP by Laplace mechanism.** We can also get tighter composition of $\varepsilon$ with relaxation to $\delta > 0$. The most straight forward way is to instantiate the privacy loss random variable of random response with $(\log(1 + p \cdot (e^{1/\sigma} - 1)), 0)$ because the dominating pair for random response is a dominating pair for the pure DP mechanism. Then, we can use the numerical composition of Pure DP PRV accountant by Gopi et al. (2021). Note that this method is agnostic to the DP mechanisms used for pure $\varepsilon$-DP. We now provide a more fine-grained privacy analysis for the Laplace mechanism. Specifically, we could compute the privacy cost of composition for the Laplace mechanism by Monte Carlo based DP accountant (Wang et al., 2023). Note that since we are dealing with scalar values, our mechanism in each iteration will be a one dimensional Laplace mechanism. Let $b$ be the scale of Laplace noise, $p$ the sub-sampling rate, and assume the sensitivity is 1, and assume we are doing composition for $T$ iterations, each iteration with sampling rate $p$. By Zhu et al. (2022) we know that the pair of distribution $(P, Q)$ dominating pair for a single dimensional Laplace mechanism, where $P$ and $Q$ are distributed according to the following probability density functions,

$$f_P = \frac{1}{2b} \exp(-|x|/b) \quad \text{and} \quad f_Q = \frac{1}{2b} \exp(-|x-1|/b).$$

Therefore, $(P, (1-p) \cdot P + p \cdot Q)$ is the dominating pair for the sub-sampled Laplace. We plug this into the standard Monte-Carlo accountant of Wang et al. (2023) (without importance sampling, see Algorithm 2 and Theorem 10 in Wang et al. (2023)) while using $10^{10}$ samples to calculate the $\delta$ at a given value of $\epsilon$. Also, using the analytical accountant explained above, we always make sure that $\mathbb{E}[\hat{\delta}_{MC}^2]$ is bounded by $10^{-8}$ (we use the fact that $\mathbb{E}[\hat{\delta}_{MC}^2]$ is bounded by $\mathbb{E}[Y^2]$ and the fact the PRV $Y$ is always bounded for Laplace mechanism). This ensures that the error of our estimation of $\delta$ is at most $10^{-8}$ with probability at least $1 - 10^{-5}$. Putting all together, for all reported values of $\varepsilon$, our $\delta$ is bounded by $10^{-5}$, with probability at least 0.99999. This privacy analysis is tighter with $\varepsilon$ is high compared to the former analysis which uses the pure DP PRV accountant. This is consistent with the intuition. As we increase the distance between the Laplace dominating pairs, the probability of sampling points from the area between the centers increases. And that is where the Laplace Mechanism is different from the Randomized Response. We present the accounting results for the Laplace method to achieve $(\varepsilon, \delta)$-DP by these two accounting methods in Table 13. Table 13 shows that the Monte Carlo based DP accountant can give tighter analysis for the Laplace mechanism for $(\varepsilon, \delta)$-DP than the pure $\varepsilon$-DP PRV method.

*Remark* A.9. We want to note a minor technicality in our privacy analysis with poisson subsampling. Zhu et al. (2022) treat the add neighboring relation and remove neighboring relation separately, because this is crucial in retaining a tight dominating pair with a closed-form expression. As shown in Zhu et al. (2022) and Lebeda et al. (2025), neither one of the add/remove neighboring relation can dominates the other one. To get the results for the standard add/remove neighboring relation for a $T$-fold composition of subsampled mechanism, we need to do the pointwise maximum of the add-only and remove-only relation. The Microsoft (2021) library only implements one of the add/remove and our randomized accounting method for subsampled Laplace mechanism only considers the add-only relation. We note that Google's-DP-Library (2020) implements DP accounting by taking maximum of the two neighboring relations under $T$ folds and provides the accounting code for both sub-sampled Gaussian and sub-sampled Laplace mechanisms. We verify that our privacy

parameters in Tables 13 and 16 to 18 do not exceed the desired $\varepsilon$ using the method of Doroshenko et al. (2022) in Google's-DP-Library (2020) with the pessimistic estimate (i.e., the upper bound).

## B  Implementation Details

We follow Malladi et al. (2023) and provide the memory-efficient version of DP-ZO in Algorithm 2. Algorithm 2 enjoys the benefit that it does not incur additional GPU memory cost compared to inference.

---

**Algorithm 2** Differentially Private-ZO (GPU memory efficient version. Adapted from Malladi et al. (2023))

---

1: Model parameters $\theta$, dataset $\mathcal{D}$, learning rate $\alpha$, perturbation scale $\phi$, random seed $s$, weight decay $\lambda$, noise scale $\sigma$, noising mechanism $\mathcal{Z}$, clipping threshold $C$, expected batch size $B$ and sampling rate $p = B/|\mathcal{D}|$. Lines with * are DP modifications.
2: **procedure** DP-ZO($(\theta, \mathcal{D}, \epsilon, \sigma, T, s, \phi, C, \alpha)$)
3:     **for** $t \in 1, \ldots T$ **do**
4:         Poisson samples $\mathcal{B}$ from dataset D with sampling rate $p$ *
5:         $\theta \leftarrow$ PerturbParameters($\theta, \phi, s$)
6:         Compute per-sample loss $\mathcal{L}_1(\theta, \mathcal{B})$*
7:         $\theta \leftarrow$ PerturbParameters($\theta, -2\phi, s$)
8:         Compute per-sample loss $\mathcal{L}_2(\theta, \mathcal{B})$*
9:         $\theta \leftarrow$ PerturbParameters($\theta, \phi, s$)
10:        Compute difference in loss $\mathcal{L} = \mathcal{L}_1 - \mathcal{L}_2$
11:        Clamp $\mathcal{L}$ between $-C$ and $C$*
12:        $g = \frac{\sum_{i \in \mathcal{B}} L + \mathcal{Z}(C, \sigma)}{B * 2\phi}$ *
13:        Reset random number generator with seed $s$
14:        **for** $\theta_i \in \theta$ **do**
15:            $z \sim \mathcal{N}(0, 1)$
16:            $\theta_i \leftarrow \theta_i - \alpha * g * z$
17:        **end for**
18:     **end for**
19: **end procedure**
20: **procedure** PerturbParameters($(\theta, \phi, s)$)
21:     Reset random number generator with seed $s$
22:     **for** $\theta_i \in \theta$ **do**
23:         $z \sim \mathcal{N}(0, 1)$
24:         $\theta_i \leftarrow \theta_i + \phi z$
25:     **end for**
26: **end procedure**

---

## C  Design Choices

Algorithm 1 outlines our DP-ZO that estimates the gradients via privatized loss value without backpropagation. In this subsection, we provide several design choices for Algorithm 1.

**Definition 2 (n-SPSA Gradient Estimator)** The n-SPSA gradient estimate averages $\nabla L_b(\theta; B)$ over $n$ randomly sampled $z$. We can write this in vector notation, dropping the normalizing constants for succinctness.

$$g_i = L(\theta + \epsilon z_i; B) - L(\theta - \epsilon z_i; B)(\text{projected gradient for each } i)$$
$$\mathbf{Z} = [z_1, z_2, ..., z_n](\text{matrix whose columns are the } z \text{ vectors})$$
$$\mathbf{g} = [g_1, g_2, ..., g_n](\text{vector of projected gradients})$$

Then the n-SPSA gradient estimate can be written as:

$$\nabla L_n(\theta; B) = \mathbf{g} \cdot Z \quad (2)$$

**How Many Gradients to be Estimated in a Model Update.** Algorithm 1 estimates the gradients once. As outlined above, SPSA can be extended to n-SPSA gradient estimator and n-SPSA can improve the performance in the non-private setting (Malladi et al., 2023). Here we discuss our design choice of why we choose $n = 1$ in Algorithm 1.

- Estimate the average. Previous work (Malladi et al., 2023) shows that averaged estimation helps the non-private setting. In a private setting, we have to privatize the gradient estimation. Here we discuss our initial design of the privatized n-SPSA gradient estimation. For the sampled batch, assuming we are adding the Gaussian noise $\mathcal{N}(0, C^2\sigma^2)$ for 1-SPSA. Then for n-SPSA, to ensure we have the same privacy cost as 1-SPSA, we need to add $\mathcal{N}(0, n \cdot C^2\sigma^2)$ to each gradient estimation and finally average the $n$ gradients. Our privacy analysis follows the $n$-fold composition of Gaussian mechanism (Corollary 3.3 in Gaussian differential privacy (Dong et al., 2019)). Our initial experiment result shows that our current analysis for n-SPSA noise addition does not make n-SPSA improve in the private setting compared to 1-SPSA. We leave the improvement in tighter analysis for private n-SPSA as future work and use 1-SPSA to conduct experiments.

**The Type of Noise for DP.** As discussed in Section 3, Algorithm 1 can be incorporated in different noise mechanisms. We focus on the Gaussian noise mechanism and the Laplace mechanism in this work. The Gaussian noise mechanism has been widely studied in previous literature both for privacy analysis and empirical performance in DP-SGD (Abadi et al., 2016; Mironov, 2017; Dong et al., 2019). The Laplace mechanism, though less studied for privacy-preserving machine learning, can provide pure DP while the Gaussian mechanism can only provide approximate DP. We have provided the privacy analysis in Section A.

## D  Experimental Details

### D.1  Datasets and Metrics

**Datasets.** Following Malladi et al. (2023), we use SST2 (Socher et al., 2013) for text classification and SQuAD (Rajpurkar et al., 2016) and DROP (Dua et al., 2019) for text generation. SST2 is a binary classification for sentiment classification based on text from movie reviews. SQuAD and DROP are question answering tasks, that given the context and question, the language model should output needed answers. Such answers are generated tokens one by one and therefore considered as generation task. In Section 4.3, we follow prior works (Yu et al., 2022; Li et al., 2022b) and run experiments on QNLI (Wang et al., 2019). QNLI is a binary classification task to determine whether the two sentences in a pair are entailment or not.

**Metrics.** Following Rajpurkar et al. (2016), we use the F1 score for text generation task. For ground truth and generation, we count the number for each different tokens (therefore ignoring the ordering of the tokens) and calculate the number of same tokens $N_{same}$ between the ground truth and the predictions. Precision is $Precision = N_{same}/N_{gt}$, where $N_{gt}$ is the total number of tokens of ground truth. Recall is $Recall = N_{same}/N_{pred}$, where $N_{pred}$ is the total number of tokens of ground truth. F1 is the harmonic mean of precision and recall $F1 = 2/(\frac{1}{precision} + \frac{1}{recall})$. We use classification accuracy for text classification as evaluation metric.

### D.2  Hyper-parameter Search

Our experiments are based on the open-source code[4] of Malladi et al. (2023). We provide the prompts we use in Table 14. In this section, we first provide several initial results for hyperparameter search on clipping threshold and finally present the hyperparameter tables. We also provide an initial study to systematically evaluate the interplay between batch size and training iterations for DP-ZO.

---

[4]https://github.com/princeton-nlp/MeZO.

Table 14: The prompts of the datasets we used for DP-ZO.

| Dataset | Type | Prompt |
|---------|------|--------|
| SQuAD | QA | Title: `<title>` |
| | | Context: `<context>` |
| | | Question: `<question>` |
| | | Answer: |
| DROP | QA | Passage: `<context>` |
| | | Question: `<question>` |
| | | Answer: |
| SST-2 | classification | `<text>` It was terrible/great |

**Different Clipping Threshold.** Li et al. (2022b); De et al. (2022) recommend small clipping $C$ threshold for DP-SGD training. For example, Li et al. (2022b) use $C = 0.1$ for training language models. We therefore study the effect of different clipping threshold and present the results in Table 15. We find that while $C = 1$ performs significantly worse, setting $C$ as 0.1, 0.05, 0.01 are within the 2% performance gap. We therefore choose $C = 0.05$.

Table 15: Different clipping C. $\sigma = 15.9$. batch size=16, 10,000 steps. $\varepsilon = 0.35$.

| | Clip=1 | Clip=0.1 | Clip=0.05 | Clip=0.01 |
|---|--------|----------|-----------|-----------|
| F1 | 66.04 | 74.26 | 76.81 | 75.39 |

**Hyper-parameter for DP-ZO (Gaussian) in Main Results.** We present the hyper-parameter for DP-ZO (Gaussian) in Table 16 and Table 17.

Table 16: Hyper-parameter search for DP-ZO in main results Table 1.

| | |
|---|---|
| $|\mathcal{D}|$ | 1000 |
| Steps $T$ | 75000 |
| Clipping $C$ | 0.05 |
| Batch size | 16 |
| $\sigma$ | 30.9 for $\varepsilon = 0.5$, 16.4 for $\varepsilon = 1$, 4.8 for $\varepsilon = 4$ |
| learning rate | [5e-6, 1e-5, 2e-5, 5e-5, 1e-4] |
| LoRA rank | 8 |
| $\phi$ | 0.01 |

Table 17: Hyperparameter search for DP-ZO with full parameter fine-tuning in Table 2.

| | |
|---|---|
| $|\mathcal{D}|$ | 1000 |
| Steps $T$ | 10000 |
| Clipping $C$ | 0.05 |
| Batch size | 16 |
| $\sigma$ | 11.47 for $\varepsilon = 0.5$, 6.08 for $\varepsilon = 1$, 1.88 for $\varepsilon = 4$ |
| learning rate | [2e-7, 5e-7, 1e-6, 2e-6, 5e-6] |
| $\phi$ | 0.001 |

**Hyper-parameter for DP-SGD.** We present the hyper-parameter search for DP-SGD in Table 18.

Table 18: Hyper-parameter search for DP-SGD in Table 2.

| | |
|---|---|
| $|\mathcal{D}|$ | 1000 |
| Steps $T$ | 200 |
| Clipping $C$ | 0.1 |
| Batch size | 64 |
| $\sigma$ | 6.60 for $\varepsilon = 0.5$, 3.59 for $\varepsilon = 1$, 1.28 for $\varepsilon = 4$ |
| learning rate | [1e-4, 2e-4, 5e-4, 1e-3, 2e-3] for LoRA fine-tuning. [1e-5, 2e-5, 5e-5, 1e-4, 2e-4, 5e-4] for Full fine-tuning. |
| LoRA rank | 8 |

**Hyper-parameter for DP-ZO (Laplace).** The hyper-parameter search for DP-ZO (Laplace) is similar to DP-ZO (Gaussian).

**Effects of Batch Size and Steps.** In Table 19 and Table 20, we did an initial study to systematically evaluate the interplay between batch size and training iterations by varying batch size in [16,32,64,128] and steps in [10000, 2000, 40000, 80000]. Similar to main results, we run 5 independent runs for each setting and compute the average of 5 runs. This set of experiments is done on OPT-13B on SQuAD dataset with LoRA fine-tuning. Table 19 and Table 20 show that increasing steps $T$ improves the performance more than increasing the batch size. We also run experiments of $T$ in [200, 400, 800, 1600] for DP-SGD (and did not observe significant improvements in DP-SGD) to ensure the fair comparison of DP-SGD and DP-ZO. Taking the computation limitation into consideration, we set $T = 75000$ and batch size BSZ=16 for main results in Table 1. We leave more investigation on the batch size and steps for DP-ZO, such as variance reduction method, as future work.

Table 19: $T = 10000$, Varying batch size BSZ.

| | BSZ=16 | BSZ=32 | BSZ=64 | BSZ=128 |
|---|---|---|---|---|
| F1 | 81.35 | 81.63 | 81.47 | 81.72 |

Table 20: Batch size=16. Varying steps $T$.

| $T$ | 10000 | 20000 | 40000 | 80000 |
|---|---|---|---|---|
| F1 | 81.35 | 81.65 | 81.42 | 82.52 |

**Computation Cost.** DP-ZO for OPT-13B models on SQuAD datasets takes around 4hrs for 10000 steps. DP-SGD for OPT-13B models on SQuAD datasets takes around 4hrs for 200 steps. When increasing $T$ or $B$ in DP-ZO, the training time scales proportionally to the scaling factor. Future work includes how to reduce the computation time of DP-ZO, e.g., by variance reduction method to improve the convergence rate.

# E   Effect of Model Size

Section 4.2 shows that DP-ZO scales to larger models and provides the results of DP-ZO for model size varying from 1.3B to 66B parameters in Table 2. Here we provide the full results of DP-ZO finetuned with LoRA at $\varepsilon = 1$, with model size ranging from 1.3B to 66B. We also include the $\varepsilon = 0$ and $\varepsilon = \infty$ baseline as a reference in Table 21.

Table 21 shows the full trend of DP-ZO with model size scaling from 1.3B to 66B, that is DP-ZO scales to larger models.

For OPT-1.3B, the gap between private and non-private baseline is 5.67. For OPT-66B, the non-private baseline is 87.49 and the gap between the private and non-private results is 3.37.

Table 21: The experiment results of DP-ZO across different model sizes. $(1, 10^{-5})$-DP.

| Model | OPT-1.3B | OPT-2.7B | OPT-6.7B | OPT-13B | OPT-30B | OPT-66B |
|---|---|---|---|---|---|---|
| $\varepsilon = 0$ | 27.20 | 29.89 | 36.48 | 46.23 | 46.53 | 48.13 |
| $\varepsilon = 1$ | $75.29_{0.90}$ | $80.34_{1.14}$ | $81.34_{1.04}$ | $82.28_{0.84}$ | $82.48_{0.83}$ | $84.12_{1.01}$ |
| $\varepsilon = \infty$ | 80.97 | 84.14 | 86.44 | 86.85 | 86.98 | 87.49 |

## F   RELATED WORK

In this section we give an overview of the broader body of work privacy preserving large language models and private zeroth-order optimization method.

**Privacy preserving large language models.** Recent studies have leveraged DP-SGD to fine-tune large language models. Li et al. (2022b) provide methods for fine-tuning large language models with DP-SGD by ghost clipping to mitigate the memory burden of per-sample gradient clipping. Yu et al. (2022) report compelling results by only updating a sparse subset of the LLMs with parameter efficient fine-tuning (PEFT) methods such as LoRA (Hu et al., 2022). He et al. (2023) leverage group-wise clipping with adaptive clipping threshold and privately fine-tune the 175 billion-parameter GPT-3. Duan et al. (2023); Li et al. (2022b) also consider private prompt tuning by adding noise to the soft prompt (Li & Liang, 2021; Lester et al., 2021). Du et al. (2023) add non-i.i.d. noise from a matrix Gaussian distribution to directly perturb embedding in the forward pass of language models. With the emergence in-context learning of large language models (Brown et al., 2020), recent works (Duan et al., 2023; Wu et al., 2024; Tang et al., 2024) study private in-context learning of large language models without fine-tuning.

**Private zeroth-order optimization.** Most recently, a concurrent work (Zhang et al., 2024a) also considers the same DP-SPSA algorithm for zeroth-order optimization. We have discussed our work and DPZero (Zhang et al., 2024a) in Section 5.

Zhang et al. (2024b) study private zeroth-order nonsmooth nonconvex optimization. Their work incorporates two zeroth-order estimators to reduce variance and samples $d$ (model dimension) i.i.d. estimators for each data point to achieve optimal dimension dependence. Zhang et al. (2024b) leverage the tree mechanism (Dwork et al., 2010; Chan et al., 2011) on disjoint data to ensure the privacy cost of the algorithm. The main focus of our work is private fine-tuning of large language models and one estimator for each batch could successfully converge in this set-up. Therefore, we only need to privatize such scalar. We leave the investigation on the private zeroth-order for more than one estimators such as the variance reduction method proposed in Zhang et al. (2024b) as future work.

Gratton et al. (2022) analyze the intrinsic privacy of the zeroth-order optimization for DP-ADMM (Huang et al., 2020) in distributed learning. Their work states that if the output of the zeroth-order method itself follows Gaussian distribution, the noise can be analyzed as the Gaussian mechanism and provide intrinsic privacy. However, this is merely stated as an assumption for lemma 1. To the best of our knowledge there is no work that proves that the zeroth-order gradient estimator can actually be analyzed as the sum of an unbiased gradient estimator and some Gaussian error term.

