# OpenReview forum: "Private Fine-tuning of Large Language Models with Zeroth-order Optimization"
_TMLR — Accepted by TMLR_

### Review · Reviewer_3HQN · 2024-09-07

**Summary Of Contributions:**

This paper studies differentially private zeroth order optimization and its application to the task of fine-tuning a pretrained LLM on a small dataset. From the current iterate $\theta$, the algorithm picks a random vector $z$ evaluates the loss on a minibatch at $\theta + \phi z$ and $\theta - \phi z$. It uses this as an estimate of the gradient at that point. This is a classic optimization procedure. Each point contributes a single scalar value to the estimate, so the paper privatizes the procedure by clipping and adding Gaussian or Laplace noise.

Compared to standard gradient methods, this estimator is simple and has lower memory overhead, as there is no backprop. Under privacy, it avoids the cost and complexity of per-gradient clipping that come with DP-SGD. Although the zeroth order estimator has high variance (that may grow with the dimension), the privacy noise under DP-SGD also grows with the dimension. So the univariate noise for privacy considered here seems very promising.

The paper evaluates the method on three NLP datasets and a variety of model sizes. Broadly, it seems to perform about as well as DP-SGD or slightly better.

**Audience:**

Yes

**Broader Impact Concerns:**

None.

**Claims And Evidence:**

No

**Requested Changes:**

The following changes will be critical to securing my recommendation:
1) All mathematical claims should be well-formed and true. Each should be accompanied by a proof or explicit reference to existing results.
2) The relationship between the submission and Zhang et al. 2024a (regarding the algorithm, publication timeline, and theoretical analysis) needs to be absolutely clear to the reader.
3) The discussion of datasets and evaluation metrics needs to provide more detail, such that a researcher from outside NLP can understand the results.

In addition, I suggest improving the presentation throughout. I feel small changes would result in a much better paper.

**Strengths And Weaknesses:**

This paper proposes a practical method for an important problem in private machine learning. It will definitely interest TMLR's audience. As with non-private zeroth order optimization, this approach presents an interesting set of tradeoffs. The empirical evaluation of utility is sufficient to convince me that the approach might be the best choice for real tasks.

There are a few serious issues with the formal statements.
1) Theorem 3.1 is not well-formed, since Algorithm 1 does not reference the parameters $\epsilon$ and $\delta$. Indeed, Algorithm 1 is not even fully specified, since it requires a "noise mechanism" as input. The privacy analyses in the appendix are completely standard, but this theorem is false.
2) Proposition 3.2 is not well-formed. As with the previous point, Algorithm 1 does not depend on any parameters $\epsilon$. Furthermore, what is the "effective rank"? What do you mean by convergence rate? How could such a convergence rate not depend on $\delta$? And, most importantly: where is the proof? Does this claim come from Zhang et al. 2024a? If so, the paper should be extremely explicit about this. Furthermore, if so the claim appears to be false because Zhang et al. 2024a analyze a different form of random perturbation.
3) Finally, a common error: Prop 2.2 is false as the conclusion does not hold for all $\epsilon \ge 0$. See [1].

If Proposition 3.2 comes from Zhang et al. 2024a, then I am deeply confused. This means the submission calls the work concurrent but uses their theoretical results? I seem to be missing something. I am also a little surprised to see Zhang et al. 2024a cited for the first time only on page 5: if their algorithm is indeed functionally the same, then it seems anyone interested in your paper would also be interested in theirs.

I did not fully understand the datasets used and experimental evaluation. What are these tasks? How do we measure accuracy? And, pardon my ignorance, how do we apply F1 scores for a generation task?

At times, I suspect the text would be difficult for an inexperienced researcher to follow. Particular examples include the introduction's Figure 1 and discussion of the algorithm and the discussion after Definition 2.4. (the definition says that $z$ is the "random perturbation." But then in this paragraph you say that "Random perturbations in zeroth-order optimization (ZO) serve as high-variance estimates of the actual gradient", which doesn't make sense since $z$ is drawn from a Gaussian. The same issue later: "... these perturbations themselves carry a privacy risk." What does that mean?)

There are a number of places where I feel the writing could be slightly improved. Examples include:
1) I feel the framing of the work as a "new methodology" and a "framework" is not quite accurate, since the algorithms seem like straightforward ways to privatize existing approaches.
2) In Section 3 paragraph "DP-ZO" we have a lot of words in italics that don't make sense to me, aside from "Poisson sampling," but even that doesn't need to be italicized.
3) The paper says that current accountants can't deal with small values of $\delta$ for subsampled Gaussian mechanism. Is this well-known? Is there a citation?
4) In Definition 2.4: what is b, what is $\phi$?
5) "Ablation" refers to the removal of something, so uses like "Ablation of DP-ZO (Gaussian) for different $n$ training samples" are not correct.

[1] Balle, Borja, and Yu-Xiang Wang. "Improving the gaussian mechanism for differential privacy: Analytical calibration and optimal denoising." International Conference on Machine Learning. PMLR, 2018.

---

> ### Author Response · Authors · 2024-11-06
>
> We thank the reviewer for their careful reading and thoughtful comments. We address all their questions below.
>
> 1. All mathematical claims should be well-formed and true.
>
> > Theorem 3.1.
>
> We have added the privacy analysis $(\varepsilon, \delta)$ per step for both subsampled Gaussian mechanism and Laplace mechanism with $\sigma$ and $p$ in the Proof Overview for Theorem 3.1. For the subsampled Gaussian mechanism, as the composition of T steps is based on the numerical composition of privacy curves, we do not provide the closed-form expression for the T steps analysis. We use such numerical composition because they provide tighter privacy accounting analysis as shown in Figure 1 in Gopi et al. 2021. We provide the T steps analysis for subsampled Laplace mechanisms in Theorem 3.1 as well as in Proof Overview. We also add the Gaussian mechanism and Laplace mechanism expression in our Algorithm 1.
>
> > Proposition 3.2
>
> Proposition 3.2 is from Zhang et al. 2024a and for DPZero from Zhang et al. 2024a and not for our DP-ZO. In our updated manuscript, we do not include the original proposition 3.2 and instead we discuss the algorithm difference between our DP-ZO and DPZero in Section 3.
>
> > privacy guarantee for Proposition 2.2.
>
> We have revised Proposition 2.2 based on Balle and Wang 2018.
>
>
> 2. Relationship with Zhang et al 2024a.
>
> We have added a paragraph in Introduction noting that “Independently and concurrently, Zhang et al. (2024a) also studied privatizing the scalar loss in Zeroth-order optimization with Gaussian noise and proposed DPZero. Our DP-ZO shares some similar motivation and design as DPZero, and there are several differences between our work and theirs. We provide a detailed discussion on DP-ZO and Zhang et al. (2024a) in Section 5.”
>
> In Section 3, we have added a paragraph discussing the algorithm difference between our DP-ZO  and DPZero in Zhang et al. 2024a including random perturbation generation strategy, data batch in each iteration and the sensitivity analysis with clipping threshold C.
>
> As the beginning of Section 5. We have added a discussion between Zhang et al. 2024a and our work.
>
> Our work is developed independently and concurrently with Zhang et al. 2024a. Since our work is not published yet, we do not include the timeline here. We are happy to provide the timeline if necessary.
>
> Besides, we note that usually the privacy analysis and popular implementation library for DP-SGD like Opacus is under add/remove DP (add or remove a datapoint for neighboring dataset) (Ponomareva et al. 2023)  and the sensitivity for DP-SGD clipping threshold C is C under add/remove DP. We follow the same add/remove DP setting and the sensitivity for DP-ZO is C as we stated in Section 3. We note that Zhang et al. 2024a analyze the sensitivity as 2C for their DPZero, which adds as much as twice noise under the add/remove DP setting. For privacy accounting, Proposition 3.2 in DPZero proof is under advanced composition that is not as tight as the numerical composition method by Gopi et al. 2021. (RenyiDP (Mironov 2017)  provides better privacy composition than advanced composition, and Gopi et al. 2021 show that their method is better than RenyiDP in their Figure 1b).  Note that (DP)-ZO takes more iterations than (DP)-SGD. With the same and tight privacy analysis method (Gopi et al. 2021) for both DP-ZO and DP-SGD, our work provides a more comprehensive understanding of the privacy-utility trade-off for DP-ZO and DP-SGD.
>
> We are also happy to engage with the reviewer feedback with our discussion.
>
> Ponomareva e al.  How to DP-fy ML: A practical guide to machine learning with differential privacy. JAIR 2023.
>
> Mironov. Renyi Differential Privacy. CSF 2017.
>
> 3. Explanation for metric and tasks.
>
> We have added a subsection for datasets and metrics in Appendix D.1.

---

> > ### Author Response · Authors · 2024-11-06
> >
> > 4. Writing.
> >
> > > “framework term”
> >
> > We have changed the term accordingly in Abstract and introduction to “method“ or “DP-ZO” such as “In this work, we study DP fine-tuning of large pretrained models with zeroth-order optimization and introduce DP-ZO.” in the second paragraph in introduction. We want to highlight that while DP-ZO is simple, its simplicity brings new insights such as DP-ZO can be instantiated with more DP mechanisms such as laplace mechanism for pure dp, while DPSGD suffers for high-dimension noises.
> > > italic.
> >
> > We have fixed these to normal fonts.
> > > Discussion after Definition 2.4 and Figure 1.
> >
> >  We have revised it into “Zeroth-order optimization (ZO) serves as high-variance estimates of the actual gradient (Liu et al., 2018), enabling optimization without the need for explicit gradient computations. While the update of model is from the random perturbation scaled by the step size where the only information from data is the step size, ZO still carries a privacy risk,  leaking information about the data (as we show later in Section 4.3).” We want to highlight that while ZO uses a scalar value from each data point in batch in each iteration, such scalar value does leak privacy, which motivates our work DP-ZO.
> > We have added in Figure 1 caption that “The only information from private data is a scalar step size for direction
> > with lower target function value”.
> >
> > > privacy accounting for small $\delta$.
> >
> > We use the composition of privacy curves (Gopi et al. 2021) for T compositions. This is based on fast fourier transformation. As noted in Wang et al 2023, the floating point issue makes it a challenge to give the tight upper bound for privacy cost. We have added the citation for Wang et al. 2023.
> > Wang et al. A randomized approach for tight privacy accounting. NeurIPS 2023.
> >
> > > Definition 2.4.
> >
> > We have revised Definition 2.4 that B is the mini batch samples from dataset D and $\phi$ is the perturbation scale.
> >
> > > Ablations.
> >
> > We have revised the wrong term 'ablations' to 'analysis'.

---

> > > ### Comment · Reviewer_3HQN · 2024-11-23
> > >
> > > Thank you for the thoughtful reply, it seems you have addressed my concerns completely.

---

> > > > ### Author Response · Authors · 2024-11-25
> > > >
> > > > We thank the reviewer for valuable suggestions and comments! We are glad that our responses could address your concern.

---

### Review · Reviewer_he8N · 2024-09-29

**Summary Of Contributions:**

While DP-SGD is the current dominant algorithm for private training and fine-tuning, it suffers from notable training time and memory overhead. This paper proposes an alternative to DP-SGD by utilizing zero-th order optimation, namely, DP-ZO. Since in ZO the gradient is estimated by the difference between two loss values, which is a scalar, it does not suffer from the curse of dimensionality thus is supposed to scale well on large models and datasets. The authors conducted extensive experiments on language tasks to demonstrate the effectiveness of DP-ZO.

**Audience:**

Yes

**Claims And Evidence:**

Yes

**Requested Changes:**

To me there are two necessary changes: (1) Theorem 3.1 (2) Expanding introduction on zero-th order optimization. Probably also need to discuss more on the comparison with DP-SGD. The benefit of DP-ZO is that you only need to add noise to a scalar, which may help improve the model utility, but the downside is you are using a less accurate gradient (an estimate) to optimize, so there is a trade-off. It is not very straightforward to me which one is clearly better than the other, which I guess depends on the application scenario.

**Strengths And Weaknesses:**

## Strengths
1. Integrating DP with ZO looks promising, as only a scalar needs to be privatized, which enjoys better scalability than DP-SGD on large models.
2. This work includes an extensive empirical study on language tasks. Evaluating privacy protection via MIA is also useful in practice.


## Weakness
1. I feel the background part about zero-th order optimization is not adequate. Readers like me who do not have knowledge in ZO may find it confusing. It seems that this algorithm still uses gradient descent, but it does not explicitly calculate the gradient, instead, the gradient is estimated by a scaled difference between function values and a random direction. For example, in Figure 1, in each iteration $\theta$  moves along one of the two random directions. If z is random, I am not clear how it is guaranteed to move towards the optimal solution.
3. I personally think the discussion of clipping threshold is important and should be included in the main body. But this is a minor point and is up to your consideration.
4. Theorem 3.1 is meaningless. I guess what you are trying to say here is that Alg 1 satisfies DP guarantee, but $\epsilon$ is not just a symbolic name, instead, it is a function of many training parameters and $\delta$. I know it might be hard to write down the explicit calculation. One alternative way is to analyze the DP guarantee of just one iteration.
6. In Table 1, DP-ZO has incremental improvements against DP-SGD, except on SST2 when $\epsilon=0.5$. The performance of DP-SGD drops a lot when $\epsilon=0.5$, but this doesn't happen on SQuAD. Can you double check this result? If nothing is wrong, can you please explain why?
7. Figure captions need to be more self-contained. For example, for Fig 2, 3, 4, the task is not even specified, readers will have no idea what the accuracy/F1 is for.
8. Found one typo (not sure if there is more): caption of Table 6

## Questions
1. it also looks to me that both $\phi$ and $\eta$ need to be very small numbers to make it work properly (is this true?) If so, does it mean that you will need more training iterations compared to DP-SGD? This might be a concern because training iterations will affect the total privacy accumulation.
5. How do you calculate the total $\epsilon$? If you are using existing libraries for DP-SGD, you need to make sure in each iteration the calculation of $\epsilon $is consistent with DP-SGD, because the rest part, e.g. subsampling and composition should be the same as DP-SGD.
3. Page 3, "...Intuitively, this scalar tells us how much better one random step is than the other..." Can you please elaborate more?
4. Page 4, "...Unfortunately, due to limitations of accounting methods, we currently cannot calculate..." What is the limitation?

---

> ### Author Response · Authors · 2024-11-06
>
> We thank the reviewer for their careful reading and thoughtful comments. We address all their questions below.
>
> 1. zeroth order optimization background.
>
>  We have revised Section 2.2 starting with “Zeroth-order optimization methods (Kiefer & Wolfowitz, 1952; Spall, 1992; Shamir, 2013; Ghadimi & Lan,2013; Nesterov & Spokoiny, 2017) use finite difference of function values to estimate gradients, instead of computing gradients in first-order methods like SGD. By evaluating the objective function values around the points x, ZO provides the step size towards the direction where the point has a lower function value; See Liu et al. (2020); Zhang et al. (2024c) for a more detailed review of Zeroth-order optimization methods.”
>
> “As noted in the Malladi et al. (2023), when $\phi\to0$, the SPSA estimate could be considered as a rank-1 reconstruction of the gradient. While SPSA only provides a scalar information from the data, interestingly, Malladi et al. (2023) show this method converges at a rate that is not catastrophically slower than SGD in fine-tuning large language models in downstream tasks. Malladi et al. (2023) reason this phenomena as a result of the Hessian of the loss exhibiting small local effective rank.”
>
> Indeed, $\phi$ should be small and we added the hyperparameter report in Table 15 and Table 16 for $\phi$ we used. We also reported the $\eta$ we used in Table 15 and Table 16, which are in the order of 1e-7 to 1e-4.
> In summary, ZO provides an estimation of gradients by choosing the random perturbation with lower target objective function value.
>
> We have added in Figure 1 caption that “The only information from private data is a scalar step size for direction
> with lower target function value”.
>
> 2. Theorem 3.1 and how to calculate total epsilon.
>
> We have added the privacy analysis $(\varepsilon, \delta)$ per step for both subsampled Gaussian mechanism and Laplace mechanism with $\sigma$ and $p$ in the Proof Overview for Theorem 3.1. For subsampled Gaussian mechanisms, as the composition of T steps is based on the numerical composition of privacy curves, we do not provide the closed-form expression for the T steps analysis. We provide the T steps analysis for subsampled Laplace mechanisms in Theorem 3.1 as well as in Proof Overview. We ensure the same setting for the subsampling and composition for DP-ZO and DP-SGD, both are poisson sampling and the numerical composition from Gopi et al. 2021.
>
> 3. More iterations in DP-ZO.
>
> We use the same accounting methods for both DP-ZO and DP-SGD. We provide the results for understanding the effect of batch size and number of iterations in DP-ZO. We find that increasing the number of iterations achieves better performance than increasing batch size in our evaluated setting in Table 18 and Table 19. We calculate the corresponding sigma with more iterations under the same privacy budget. We note that indeed we cannot increase the number of iterations in DP-ZO to arbitrary numbers, but in the evaluated number of iterations in Appendix D.2,  the performance of DP-ZO improved.
>
> 4. Results for $\varepsilon=0.5$ for SST2.
>
> We have checked and rerun, the result for SST2 is as it is. Note that we average 5 runs with different seeds for models and optimizers for Table 1. As we discussed in Section 4, the few-shot learning setting is challenging due to the privacy requirement for a limited number of samples while the few-shot learning setting is a practical scenario. In this challenging setting, the result of DP-SGD includes high variance. While we use the same privacy-related parameters for the three tasks, this indicates that the tasks may exhibit different levels of robustness to the added noise.
>
> 5. Discussion with DP-SGD.
>
> We present the comparison for utility-privacy trade-off in Table 1,2,3,4,9,10. We also discuss the computation benefits of DP-ZO in Section 5. At the end of Section 5, we outline the challenge for DP-ZO when there is a utility gap between ZO and SGD.

---

> > ### Author Response · Authors · 2024-11-06
> >
> > 6. Writings.
> >
> > > “Page 3, this scalar tells us how much better one random step is than the other...”.
> >
> > zeroth-order optimization uses finite difference of function values to estimate gradients. If the target function value is lower, such random perturbation will be chosen as updates.
> > > figure 2,3,4 captions.
> >
> > We have updated Figure 3 and Figure 4 specifying SQuAd tasks. We have added Figure 2 caption that F1 is for SQuAD and DROP, and accuracy is for SST2.
> > > typo in Table 6.
> >
> > We have fixed it. We will also do proofreading.
> > > privacy accounting for small $\delta$.
> >
> >  We use the composition of privacy curves (Gopi et al. 2021) for T compositions. This is based on fast fourier transformation. As noted in Wang et al 2023, the floating point issue makes it a challenge to give the tight upper bound for privacy cost. We have added the citation for Wang et al. 2023
> > Wang et al. A randomized approach for tight privacy accounting. NeurIPS 2023.
> >
> > > discussion for clipping threshold.
> >
> > Our current version stays with 12 pages for the main body and puts discussion clipping threshold in the appendix. We will improve the manuscript outline if with more pages.

---

> > > ### Comment · Reviewer_he8N · 2024-11-16
> > >
> > > Thanks. Most of my concerns are addressed properly, and I don't have more questions for now.

---

> > > > ### Author Response · Authors · 2024-11-17
> > > >
> > > > We thank the reviewer for the feedback! We appreciate your time, comments, and valuable suggestions.

---

### Review · Reviewer_7MQH · 2024-10-23

**Summary Of Contributions:**

This paper studies differentially private (DP) zeroth order (ZO) optimization in the context of fewshot finetuning of Large Language Models (LLMs). It is noted that adding DP to ZO optimization has minimal computational overhead as one only needs to privatize the directional derivative along some randomly chosen direction. Note that DP-SGD requires privatizing the full gradient and it involves per-sample gradient clipping. Although the paper is claiming to have identified this idea, it is already known (see weakness). Inspired by this method authors propose an algorithm, DP-ZO, and attempts (see weakness) to provide privacy and convergence rate analysis for it. Rest (most) of the paper provides extensive experimental evidence to argue that DP-ZO has several advantages over the standard DP-SGD, including (1) comparable performance in fewshot finetuning, (2) considerably smaller memory consumption, and (3) practicality of exact epsilon-DP through DP-ZO which is impractical to achieve through DP-SGD due to additional dimension factors (see weakness). Although a few of these results are already known in some settings, I consider the comprehensive empirical results as useful and major contribution of this paper.

**Audience:**

Yes

**Claims And Evidence:**

No

**Requested Changes:**

Please address my concerns in weaknesses and minor comments.

**Strengths And Weaknesses:**

Strengths:

1. Extensive experimental results testing various hypotheses in different settings (models, datasets, privacy levels, dataset size, etc.), including experiments showing the competitiveness of DP-ZO and its memory efficiency.

    a. Papers also show that gap between DP-ZO and DP-SGD reduces as the model size increases

    b. I also very much appreciate that the authors include a very useful negative results (Table 9) which shows that the gap between DP-ZO and DP-SGD increase with number of samples.

2. Paper identifies that “exact” epsilon-DP which incurs additional dimension dependent terms in DP-SGD (and hence impractical) is practical in DP-ZO (also see weakness).
3. Authors also use Membership Inference Attacks (MIA) to analyze the privacy leakage of DP-ZO and conclude that inherent noise in sampling the direction for the derivative boosts the empirical privacy of DP-ZO over DP-SGD.
4. Authors also argue in detail that scaling DP-ZO to larger models though sophisticated model training paradigms (e.g. model/data parallelism) is easier to implement than DP-SGD which requires per-sample gradient clipping instead of per-sample directional derivative clipping.

Weaknesses:

1. Theory is wrong and incomplete: Although authors claim to provide privacy analysis and convergence rate of their method but it seems incomplete and wrong. Some issues are noted below

    a. Theorem 3.1 statement appears incomplete as algorithm hyper parameters values are not defined.

    b. Number of iterations T does not appear in the privacy analysis of Gaussian mechanism?

    c. Clipping factor C does not appear in the analysis of Laplace mechanism

    d. Proof contains too much hand-waving. Proof by words (jargon) instead of more precise mathematical statements. What are PRV and “dominating pair”?

    e. It is also quite peculiar that proof (Appendix A) computes precise values for \delta which never appeared in the theorem statement.

    f. There are additional dimension factors appearing in zeroth order optimization. So it is not clear whether Laplace mechanism would be practical in all use cases of DP-ZO such as full finetuning or training from scratch. Making theory more precise would help with this confusion.

    g. Proposition 3.2 is given without any proof. Authors allude that this follows from Malladi et al. (2023). However, this prior work precisely proves it for non-DP ZO algorithm under certain conditions including taking limit of \phi. However, it is not at all clear how the same applies to DP-ZO (with non-zero \phi) which adds clipping and noise perturbation. One concern is that clipping factor or noise may be dimension dependent and lead to worse convergence rate for DP-ZO.

    h. “In the non-private setting where the adaptation between the pretrained model and the fine-tuning dataset has low rank (Hu et al., 2022), as in fine-tuning large language models, Malladi et al. (2023) show this method converges at a rate that is not catastrophically slower than SGD fine-tuning.” This is incorrect reading of (Malladi et al., 2023) which assumes low rank Hessian of the objective. This is clearly different from assuming low rank in LoRA finetuning where of Hessian could still have large rank.

2. Prior work not properly attributed. Authors’ claim that (Zhang et al., 2024a) is a concurrent and independent work is hard to justify as this paper appeared in October 2023. (Zhang et al., 2024a) seems to have already made the observation that adding DP to ZO is computationally cheap and proved it theoretically. It is also worrisome that authors do not mention this paper till page 5 and only discuss their contribution in page 12. According to standard academic conventions, it is more appropriate to discuss very relevant prior work in the introduction and only claim the novel contributions (see strengths) as original as opposed to “new methodology DP-ZO” (on page 1).

3. Why are different settings (models, datasets, etc) chosen for different experiments. E.g. Table 7 vs 9, Table 1 vs 2. Usually arbitrary experimental choices like these increases the chances of wrong conclusion by sheer randomness.

   a. How does DP-ZO compare with MeZO and DP-SGD when doing full fine tuning on these 3 datasets instead of QNLI?

4. Paper is not well written mainly due to many undefined jargon and terminology. Sometimes sentences are hard to parse.

    a. “empirical privacy analysis is still much higher than random guess” What does this mean?

    b. “Random perturbations in zeroth-order optimization (ZO) serve as high-variance estimates of the actual gradient, enabling optimization without the need for explicit gradient computations. However, these perturbations themselves carry a privacy risk” What does this mean?

    c. l_p sensitivity is never defined even though used in words in the proof of Theorem 3.1

    d. “add-remove DP” in page 4 is never defined or referenced

    e. “privately fine-tuning downstream tasks in the few-shot setting will be more aligned with real-world use cases.”: It is not clear what authors meant by this line and paragraph.

   f. “clipping are known to degrade performance,”: Please add a reference for this claim? Can it be proved empirically?

5. Although authors provide extensive results for the memory usage, paper is missing comparison overall run time and number of iterations used by various methods.

These suggest that the manuscript is not yet ready for publication.

Minor comments:
1. Table 9: do you have similar gap between ZO and SGD for non-private eps=\infty?
2. Table 2: Why does gap between DP-ZO and DP-SGD have opposite trends across LoRA and full finetuning?
3. Figure 1 is confusing. Also there seems to be some violation of format spacing between caption and main text.
5. Sec 5.1: [1] suggested similar idea of communicating only seeds and direction derivatives to save computation in federated learning
6. Typo: “activatiosn”

[1] Li, Z., Ying, B., Liu, Z., & Yang, H. (2024). Achieving Dimension-Free Communication in Federated Learning via Zeroth-Order Optimization. arXiv preprint arXiv:2405.15861.

---

> ### Author Response · Authors · 2024-11-06
>
> We thank the reviewer for their careful reading and thoughtful comments. We address all their questions below.
>
> 1.Theorem 3.1.
> > privacy parameters and analysis for T iterations.
>
> We have added the privacy analysis $(\varepsilon, \delta)$ per step for both subsampled Gaussian mechanism and Laplace mechanism with $\sigma$ and $p$ in the Proof Overview for Theorem 3.1. For the subsampled Gaussian mechanism, as the composition of T steps is based on the numerical composition of privacy curves, we do not provide the closed-form expression for the T steps analysis. We use such numerical composition because they provide tighter privacy accounting analysis than others as shown in Figure 1 in Gopi et al. 2021. We provide the T steps analysis for subsampled Laplace mechanisms in Theorem 3.1 as well as in Proof Overview. We also add the Gaussian mechanism and Laplace mechanism expression in our Algorithm 1.
>
> > clipping factor.
>
> We have fixed the typo and added $C$.
>
> > “PRV and dominating pairs” and $\delta$ as a function of $\varepsilon$.
>
> We have added necessary definitions and propositions in Appendix A (A.3-A.8). Dominating pairs is defined in Definitions A.6 and Privacy Loss Random Variable is in Defintion A.8. We added the Definition A.5 for optimal privacy curves. We follow prior works (Sommer et al. (2019); Koskela et al. (2020); Gopi et al. (2021), Zhu et al. (2022); Wang et al. (2023)) as the composition of optimal privacy curves could give a tighter privacy accountant.
>
> 2. Proposition 3.2 and the “low rank hessian of objective”
>
> Proposition 3.2 is from Zhang et al. 2024a and for DPZero from Zhang et al. 2024a and not for our DP-ZO. In our updated manuscript, we do not include the original proposition 3.2 and instead we discuss the algorithm difference between our DP-ZO and DPZero in Section 3.
>
> We have revised Section 2.2 as “While SPSA only provides a scalar information from the data, interestingly, Malladi et al. (2023) show this method converges at a rate that is not catastrophically slower than SGD in fine-tuning large language models in downstream tasks. Malladi et al. (2023) reason this phenomena as a result of the Hessian of the loss exhibiting small local effective rank.”
>
> 3. Relationship with Zhang et al 2024a.
>
> We have added a paragraph in Introduction noting that “Independently and concurrently, Zhang et al. (2024a) also studied privatizing the scalar loss in Zeroth-order optimization with Gaussian noise and proposed DPZero. Our DP-ZO shares some similar motivation and design as DPZero, and there are several differences between our work and theirs. We provide a detailed discussion on DP-ZO and Zhang et al. (2024a) in Section 5.”
>
> In Section 3, we have added a paragraph discussing the algorithm difference between our DP-ZO  and DPZero in Zhang et al. 2024a including random perturbation generation strategy, data batch in each iteration and the sensitivity analysis with clipping threshold C.
>
> As the beginning of Section 5. We have added a discussion between Zhang et al. 2024a and our work.
>
> Our work is developed independently and concurrently with Zhang et al. 2024a. Since our work is not published yet, we do not include the timeline here. We are happy to provide the timeline if necessary.
>
> Besides, we note that usually the privacy analysis and popular implementation library for DP-SGD like Opacus is under add/remove DP (add or remove a datapoint for neighboring dataset) (Ponomareva et al. 2023)  and the sensitivity for DP-SGD clipping threshold C is C under add/remove DP. We follow the same add/remove DP setting and the sensitivity for DP-ZO is C as we stated in Section 3. We note that Zhang et al. 2024a analyze the sensitivity as 2C for their DPZero, which adds as much as twice noise under the add/remove DP setting. For privacy accounting, the proof in DPZero is under advanced composition that is not as tight as the numerical composition method by Gopi et al. 2021. (RenyiDP (Mironov 2017)  provides better privacy composition than advanced composition, and Gopi et al. 2021 show that their method is better than RenyiDP in their Figure 1b).  Note that (DP)-ZO takes more iterations than (DP)-SGD (in our Table 15 and 17) ,with the same and tight privacy analysis method (Gopi et al. 2021) for both DP-ZO and DP-SGD, our work provides a more comprehensive understanding of the privacy-utility trade-off for DP-ZO and DP-SGD.
>
> We are also happy to engage with the reviewer feedback with our discussion.
>
> Ponomareva et al.  How to DP-fy ML: A practical guide to machine learning with differential privacy. JAIR 2023.
>
> Mironov. Renyi Differential Privacy. CSF 2017.

---

> > ### Author Response · Authors · 2024-11-06
> >
> > 4. Table 9. Gap between ZO and SGD. and results for datasets in Table 1.
> >
> > Due to limited computation resources and response time window, we provide the results for non-private results on roberta-base QNLI for training examples n=1000 and n=104743. SGD achieves 84.00 for n=1000 and 89.49 for n=104743. ZO achieves 79.85 for n=1000 and 79.99 for n=104743. This result is consistent with the result in Table 7. The non-private results of SGD and ZO in Table 7 and Table 9 indicate that the utility challenge for ZO under this case and therefore for the private version DP-ZO there is also utility challenge in such case.
> >
> > We also provide similar observations for OPT-13b models with LoRA fine-tuning on SST2 for n=66849, i.e., full training dataset. SGD achieves $96.33$ and DP-SGD achieves $95.64$ for $\varepsilon=1$. ZO achieves $94.84$ and DP-ZO achievers $93.00$ for $\varepsilon=1$. Comparing this results and Table 1, this is consistent with our observations on Roberta-base on QNLI task. We will add this result in our updated manuscript.
> >
> > 5. Laplace in practical use cases of DP-ZO such as full finetuning and training from scratch.
> >
> > We studied the fine-tuning large language models in this work. As discussed in Gao et al. 2021, finetuning is the practical scenario.  Malladi et al. 2023 show that how ZO converges at a rate that is not catastrophically slower
> > than SGD in fine-tuning large language models in downstream tasks. We acknowledge that training from scratch is still a challenging task for ZO. As we see in the limitations of DP-ZO and ZO in Table 7, the performance of DP-ZO is limited by ZO. DP-ZO with Laplace is still a challenging task when training from scratch. We have added a discussion for DP-SGD vs. DP-ZO at the last paragraph in Section 5.
> >
> > Due to limited time, we do not provide the results for DP-ZO laplace for full parameter finetuning. We anticipate that at least DP-ZO (Laplace) is better than DP-SGD (laplace), this is because the full parameter finetuning has more trainable parameters, and DP-SGD (laplace) adds laplace noise to all parameters, which would not be better than DP-SGD with Laplace by LoRA. In contrast, if ZO full finetuning is applicable, such as the case we show in Table 1, we anticipate that DP-ZO (laplace) can still optimize as we still only add noise to a scalar value.
> >
> > 6. Table 2 full and LoRA.
> >
> > Prior work has observed that DP-SGD may be better with dimensionality reduction such as LoRA (Kurakin et al. 2023). Similarly, for ZO, in a very high-dimensional space the likelihood of randomly perturbing good parameters is low, so ZO generally prefers dimensionality reduction techniques. This is identified in the Malladi et al. 2023 Table 20, where the 66B-parameter model with full parameter fine-tuning needs further tuning to successfully optimize in the evaluated experimental set-up and requires parameter efficient methods. DP-ZO inherits this behavior, so DP-ZO prefers dimensionality reduction techniques such as LoRA.
> >
> > Kurakin et al.. Harnessing large-language models to generate private synthetic text. arxiv 2023.
> >
> > 7. Computation time and number of iterations.
> >
> > We include the results for understanding the effect of batch size and number of iterations in DP-ZO in Appendix D. We find that increasing the number of iterations achieves better performance than increasing batch size in our evaluated setting in Table 18 and Table 19. We have a rough overall computation time estimation in the last paragraph in Appendix D. We provide this rough estimation in Appendix.  We currently stay with this outline to stay with 12 pages for the main body.

---

> > > ### Author Response · Authors · 2024-11-06
> > >
> > > 8. Writing.
> > >
> > > > “empirical privacy analysis is still much higher than random guess”.
> > >
> > > We use membership inference attacks to measure empirical privacy leakage and have revised accordingly “such empirical privacy leakage estimated by membership inference attacks (Shokri et al., 2017; Panda et al., 2024b) is still much higher than random guess.”
> > >
> > > > “Random perturbations in zeroth-order optimization (ZO) serve as high-variance estimates of the actual gradient,
> > > enabling optimization without the need for explicit gradient computations. However, these perturbations themselves
> > > carry a privacy risk”.
> > >
> > > We have revised it into “Zeroth-order optimization (ZO) serves as high-variance estimates of the actual gradient (Liu et al., 2018), enabling optimization without the need for explicit gradient computations. While the update of model is from the random perturbation scaled by the step size where the only information from data is the step size, ZO still carries a privacy risk,  leaking information about the data (as we show later in Section 4.3).”
> > >
> > > We want to highlight that while ZO uses a scalar value from each data point in batch in each iteration, such scalar value does leak privacy, which motivates our work DP-ZO.
> > >
> > > > l_p sensitivity.
> > >
> > > We have added l_p sensitivity before Proposition 2.2
> > >
> > > > add/remove DP.
> > >
> > > We have added a footnote for add/remove DP in Definition 2.1.
> > >
> > > > “privately fine-tuning downstream tasks in the few-shot setting will be more aligned with real-world use cases.”
> > >
> > > As stated in Gao et al. 2021, in practical scenarios, the available data might be limited due to data availability and additional annotation efforts. We have added the citations in the updated manuscript.
> > >
> > > Gao et al. Making Pre-trained Language Models Better Few-shot Learners, ACL 2021.
> > >
> > > > “clipping are known to degrade performance,”
> > >
> > > We have revised it as “small threshold like C = 0.1 for per-example clipping makes gradient estimator biased and loss convergence issues arise in σ = 0 for DP-SGD (Andrew et al., 2021; Chen et al., 2020; De et al., 2022; Bu et al., 2023b),” Bu et al. 2023b provided the such empirical validation in Figure 1, where the loss trajectory is different for $\varepsilon=\infty, \sigma=0$ in DP-SGD and the standard SGD.
> > >
> > > > typo.
> > >
> > > We have fixed typos. We will also proofreading for the manuscript.
> > >
> > > > communicating seeds only.
> > >
> > > We have added the reference in Section 5 discussion.
> > >
> > > > Figure 1.
> > >
> > > We have added in Figure 1 caption that “The only information from private data is a scalar step size for direction with lower target function value”. The main body text and Figure 1 caption are visually separable now.

---

> > > > ### Comment · Reviewer_7MQH · 2024-12-07
> > > > **Thank you for response**
> > > >
> > > > Dear authors
> > > >
> > > > Thank you for your response to my review and updating the manuscript. You have addressed most of my comments. My major remaining concern is the fact that the proof of the remaining theorem is still hard to parse for a non-expert. Additionally it is appealing to numerical arguments while not presenting enough details.
> > > >
> > > > Regards
> > > >
> > > > Reviewer 7MQH

---

> ### Author Response · Authors · 2024-12-10
>
> We thank the reviewer for the thoughtful comments! We are glad that we have addressed most of your comments.
>
> For the privacy analysis proof, the per-step analysis for Gaussian and Laplace mechanism has the closed form expression as in Section 3, and Appendix A. We use the Hockey-stick Divergence for per-step analysis of the Gaussian mechanism because it is tight [Zhu et al. 2022] (our Definition A.3, Lemma A.4, Definition A.5).
>
> We now give a few more background for the composition of DP mechanisms within privacy variables and privacy loss distribution.
>
> Analyzing the privacy guarantees of a mechanism can be done via worst-case inputs $(x_0, x_1)$ to a mechanism M leading to a pair of worst-case distributions [3]. Meiser and Mohammadi [2] introduce a novel method to approximately compose the privacy curves based on the discretized version of privacy loss random variables [4] , whose distribution is called privacy loss distribution (PLD). Meiser and Mohammadi [2] provide the worst-case pair of distributions for basic mechanisms such as Gaussian mechanism and Laplace mechanism. Sommer et al. [5] derive the exact analytical and closed formula for the Gaussian mechanism.
>
> Zhu et al.[1] formalize a rigorous notion of the "worst-case" PLD for DP mechanism under the name dominating PLDs. Zhu et al.[1] prove that any privacy mechanism has a tightly dominating pair of distributions and provide the dominating pairs for basic privacy mechanisms such as Gaussian mechanism and Laplace mechanism. Zhu et al. also provide an accounting method for computing tight DP composition bounds when the analytical expression for the characteristic function of the PLDs is known. We therefore introduce the Definition for Dominating Pairs in Definition A.6. Based on Definition A.6, along with Definition A.3, we can introduce the privacy loss random variable to convert the hockey-stick divergence to an efficiently computable form, therefore we introduce the Definition A.8. Compared to the current outline, Proposition A.7 can be stated after Definition A.8 so that Definition A.3, A.5, A.6, A.8 can first outline how we do the privacy analysis for given the privacy dominating pairs. Note that Zhu et al. have provided the dominating pairs for basic privacy mechanisms such as Gaussian mechanism and Laplace mechanism and therefore we could directly use.
>
> Next we can introduce Proposition A.7 for the privacy amplification by poisson sampling within the expression by dominating pairs.
>
> Given a dominating pair, Zhu et al.[1] show that if the characteristic function of the PLDs has an analytical expression, for example, Gaussian mechanism, the tight DP composition bounds can be computed in O(1) times. However, without the closed form, such as the sub-sampled Gaussian and sub-sampled Laplace, numerical accounting methods are proposed to approximate the integral composition formula. Koskela et al. [6] propose to speed up discrete convolution for numerical approximation of the integral composition formula by the fast Fourier transform algorithm. The error analysis and improved algorithms in this direction are studied in [6, 7, 8, 9]. Such numerical methods provide better privacy composition than other analytic composition methods such as RenyiDP and advanced composition theorem as shown in Zhu et al.[1] Figure 3(a) (Gaussian mechanism, no subsampling) and Gopi et al. [9] Figure 1 (subsampled Gaussian mechanism).
>
> As we show through the initial study of increasing batch size vs increasing number of steps, we find that increasing the number of iterations achieves better performance than increasing batch size in our evaluated setting in Table 18 and Table 19. Therefore we use the subsampled DP mechanism, i.e., subsampled Gaussian mechanism and subsampled Laplace mechanism, to enable a small batch size at each step instead of full training set at each step. Therefore, the privacy mechanism in our experiments are subsampled DP mechanisms and privacy analysis for T steps in our experiments is done by the numerical composition methods.

---

> ### Author Response · Authors · 2024-12-10
>
> There are multiple open-source implementations for numerically accurate composition including [10, 11] that are widely used by other DP-SGD works [12, 13] to enjoy a better privacy amplification by subsampling benefits by these tight numerical composition methods. In our experiments, we use [10] to conduct the DP analysis for both DP-ZO and DP-SGD. We also report the privacy analysis related parameters (subsampling $p=B/|D|$, steps $T$, noise multiplier $\sigma$ ) in Appendix (Table 11, 12, 15, 16, 17 ) to enhance the reproducibility of our work.
>
> For example, in Table 15 for DP-ZO, we use $\sigma=16.4$ for $\varepsilon=1$, with batch size $B=16$ for $T=75000$ steps. Total data size is $|D|=1000$. Following the example in https://github.com/microsoft/prv_accountant?tab=readme-ov-file#example, we can compute $\varepsilon$ at $\delta=1e-5$ as
> ```
> sample_rate = 16/1000
> sigma = 16.4
> step = 75000
>
> delta = 1/(1e5)
>
> mech = PoissonSubsampledGaussianMechanism(
>     noise_multiplier=sigma,
>     sampling_probability=sample_rate,
> )
> accountant = PRVAccountant(
>     prvs=[mech],
>     max_self_compositions=[step+10],
>     eps_error=0.001,
>     delta_error=1e-10
> )
> eps_low, eps_est_init, eps_up = accountant.compute_epsilon(delta=delta, num_self_compositions=[step])
> print( eps_low, eps_est_init, eps_up)
> ```
> We get $eps_{up}=0.9988576386495023$, that is the upper bound of the T composition, does not exceed 1.
>
> We ensure using the same composition technique [10] for DP-ZO and DP-SGD in our experiments.
>
> [1] Yuqing Zhu Jinshuo Dong, and Yu-Xiang Wang. Optimal Accounting of Differential Privacy via Characteristic Function. AISTATS 2022.
>
> [2] Sebastian Meiser and Esfandiar Mohammadi. Tight on Budget?: Tight Bounds for r-Fold Approximate Differential Privacy. CCS 2018.
>
> [3] Cynthia Dwork and Aaron Roth. The Algorithmic Foundations of Differential Privacy. Foundations and Trends in Theoretical Computer Science, 2014.
>
> [4] Cynthia Dwork and Guy N. Rothblum. Concentrated Differential Privacy. arxiv preprint 1603.01887, 2016.
>
> [5] David M. Sommer,  Sebastian Meiser, and Esfandiar Mohammadi. Privacy loss classes: The central limit theorem in differential privacy. PETS, 2019
>
> [6] Antti Koskela, Joonas Jälkö, and Antti Honkela, Computing Tight Differential Privacy Guarantees Using FFT. AISTATS 2020.
>
> [7] Antti Koskela and Antti Honkela. Computing differential privacy guarantees for heterogeneous compositions using fft. arXiv preprint arXiv:2102.12412, 2021.
>
> [8]  Antti Koskela, Joonas Jälkö, Lukas Prediger, and Antti Honkela. Tight differential privacy for discrete-valued mechanisms and for the subsampled gaussian mechanism using fft. AISTATS 2021.
>
> [9] Sivakanth Gopi, Yin Tat Lee, and Lukas Wutschitz. Numerical composition of differential privacy. NeurIPS, 2021.
>
> [10] Microsoft. https://github.com/microsoft/prv_accountant
>
> [11] Google DP library. https://github.com/google/differential-privacy.
>
> [12] Yu et al. Differentially Private Fine-tuning of Language Models. ICLR 2022.
>
> [13] Chua et al. How Private are DP-SGD Implementations? ICML 2024.

---

> ### Comment · Reviewer_7MQH · 2024-12-11
> **Accounting for Gaussian mechanism**
>
> Thank you for sharing the details of the accounting process. Please include them in next revision. May I ask why tables like Table 11, 12 are not present for the Gaussian mechanism?

---

> > ### Author Response · Authors · 2024-12-15
> >
> > Thank you for your response! We will include this accounting detail in our earlier response in our next revision.
> >
> > Table 11, 12 are for Laplace mechanism and Table 15, 16, 17 are for Gaussian mechanism. Table 3, 8, 10 are results by Laplace mechanism.Specifically, Table 3,8 are results for Laplace mechanism for pure-$\varepsilon$-DP and Table 10 is result for Laplace mechanism for  approximate $(\varepsilon, \delta)$-DP. For pure-DP, our privacy analysis gives $(T\times \log(1 + p · (e^{1/σ} − 1)), 0)$-DP, for given privacy parameters noise multiplier $\sigma$, subsampling rate $p$, steps $T$. We report these parameters $\sigma, p, T$ for each $\varepsilon$ we use for pure-DP in Table 11. Table 12 is for Laplace mechanism for $(\varepsilon,\delta)$ and such privacy accounting technique could be done either by 1) using the proposed method by Wang et al. [14] based on dominating pair for Laplace mechanism 2) using the dominating pair of pure-DP private random variable (similar to the method in Tang et al.[15]). As in Table 12 we can see that the first method provide better privacy accounting result, this is because the first method is based on mechanism specific analysis and is therefore better.
> >
> > We will add the description in the caption of Table 11, 12, 15, 16, 17 to clarify the their corresponding results tables.
> >
> > [14] Jiachen T Wang, Saeed Mahloujifar, Tong Wu, Ruoxi Jia, and Prateek Mittal. A randomized approach for tight privacy accounting. NeurIPS, 2023.
> >
> > [15] Xinyu Tang, Richard Shin, Huseyin A Inan, Andre Manoel, Fatemehsadat Mireshghallah, Zinan Lin, Sivakanth Gopi, Janardhan Kulkarni, and Robert Sim. Privacy-preserving in-context learning with differentially private few-shot generation. ICLR, 2024.

---

### Decision · Action_Editor_Ccvq · 2025-01-02

**Recommendation:** Accept with minor revision

**Comment:**

The authors have taken into account reviewers' comments during the review process and as a result the paper has improved. I think it is almost ready to be accepted as is, however, I think still a minor revision is required to address some remaining reviewers' comments.

Please still carefully go over the statement of Theorem 3.1, the "proof outline" and the related material in the appendix (also requested by reviewer 7MQH). I think that in Proposition A.7, the dominating pair of distributions you state is the dominating pair only in the "add" case. Rigorously speaking, you should consider both add and remove cases, see Theorem 11 by [Zhu et al., 2022](https://proceedings.mlr.press/v151/zhu22c/zhu22c.pdf). I believe the $\varepsilon$-values you report are correct: as far as I know PRV accountant implements only one of the cases, however it seems to always give the upper bound though it hasn't been rigorously shown that it gives the upper bounds for add/remove (see conjecture in [Lebeda et al., 20249(https://arxiv.org/pdf/2405.20769)). So, given the experimental nature of this paper, I think its ok to rely on the $\varepsilon$-values given by the PRV accountant and simply fix the theoretical part (e.g., mention both cases, and that you need to take the maximum of the two resulting $\delta$-values). Also, you could add to the discussion of Appendix A that the privacy loss random variables add up in case of compositions.

Also, I would request to still carefully go through the writing. Please go carefully through examples in reviewers comments. Few examples of strang sentences I found:
- "still causes OOM error for sequence length equals to 2048"
- "In fact, DP-ZO incurs nearly no additional memory cost than ZO"
- "DP-ZO ss flexible enough to be extended..."

In Definition 2.1 you write $D,D' \in \mathcal{D}^n$ however you use add/remove neighbourhood relation. Perhaps consider changing the notation or further clarifying the explanation "...if they differ in exactly one record by adding or removing one record".

Other remarks:

- you mention "Unfortunately, due to limitations of accounting methods, there are challenges for calculating the tight privacy of composition of sub-sampled Gaussian mechanism for values of $\delta$ smaller than $10^{-10}$ (Wang et al., 2023)." Notice that there are method for particularly addressing small $\delta$-values, see e.g. [Alghamdi et al., 2022](https://arxiv.org/pdf/2208.09595).

- You are using randomized privacy accounting for $(\varepsilon,\delta)$-DP analysis of the Laplace mechanism. Notice that you could also use FFT-based accounting. For example, the PLD for one of the cases (add / remove) is given in Appendix A.3 of [Sommer et al., 2020](https://eprint.iacr.org/2018/820.pdf).

**Audience:**

Yes, the paper fits very well to TMLR.

**Claims And Evidence:**

The paper presents DP-ZO, a novel approach for fine-tuning LLMs with DP through zeroth-order optimization. Unlike traditional DP-SGD methods, which rely on noisy gradient updates and can be memory-intensive, DP-ZO adds noise only to a scalar step size derived from differences in loss values, making it more memory-efficient and scalable. It is experimentally shown that the method achieves strong privacy-utility tradeoffs, with performance sometimes on par with DP-SGD. DP-ZO simplifies implementation for large models and scales effectively without the gradient clipping required by DP-SGD. The method also enables pure $\varepsilon$-DP.

This topic has become an active research topic recently, and there is a related ICML 2024 paper [Zhang et al. "DPZero: Private Fine-Tuning of Language Models without Backpropagation"](https://arxiv.org/pdf/2310.09639), which proposes a method (their Alg. 2) that is almost equivalent to the method proposed in this paper when using Gaussian noise (different scaling used for the gradient approximation). This paper generalized the method by Zhang et al. by considering applying also the Laplace mechanism to the step length. The paper has also extensive experimental comparisons and therefore complements the existing research on this topic. There are interesting experimental observations such as the comparison to DP-SGD when the dataset size varies (Table 9: it remains as an open problem how to match the utility of DP-SGD for large data).

---

> ### Author Response · Authors · 2025-01-28
> **Camera-ready version**
>
> We thank all reviewers and the Action Editor for the constructive feedback, which has helped improve the quality of our paper.
>
> We have uploaded the camera-ready version. In the latest revision, we have updated:
>
> > Details for DP accounting.
>
> For Theorem 3.1, Proposition A.7 and the single-step analysis for Gaussian mechanism in Appendix A.1, we have revised the $(\varepsilon, \delta)$-DP analysis for add/remove neighboring is by taking the pointwise maximum of the add-only and remove-only relation.
>
> We add the privacy accounting details in Appendix A.1 noting that when we use the Microsoft PRV accountant, we ensure the upper bound by the Microsoft PRV accountant will not exceed the desired $\varepsilon$ level.
>
> We also added a remark as a final paragraph in Appendix A.1 noting that the Microsoft PRV accountant and our method for subsampled Laplace mechanism for (\epsilon, \delta)-DP only consider one of the add/remove neighboring relation. We further note that the google DP library implements DP accounting method by taking maximum of the two neighboring relations under T folds and provides the accounting code for sub-sampled Gaussian and sub-sampled Laplace mechanisms. We verify that our privacy parameters for subsampled Gaussian and subsampled Laplace mechanisms in Tables 13 and 16 to 18 do not exceed the desired ε using the method of Doroshenko et al. (2022) in Google’s-DP-Library (2020) with the pessimistic estimate (i.e., the upper bound) under (\varepsilon, \delta)-DP.
>
> We motivate the pure $\varepsilon$-DP by Laplace mechanism as the stronger privacy notion compared to ($\varepsilon,\delta$)-DP in Section 3.
>
> > Writing quality.
>
> We have incorporated the feedback from reviewers and AE, and worked to improve clarity and avoid typos.